# NAD(H) homeostasis underlies host protection mediated by glycolytic myeloid cells in tuberculosis

Hayden T. Pacl [1], Krishna C. Chinta [1], Vineel P. Reddy [1], Sajid Nadeem[1], Ritesh R. Sevalkar[1], Kievershen Nargan[2], Kapongo Lumamba[2], Threnesan Naidoo[2,3], Joel N. Glasgow [1], Anupam Agarwal [4] ✉ & Adrie J. C. Steyn [1,2,5] ✉

*Mycobacterium tuberculosis* (*Mtb*) disrupts glycolytic flux in infected myeloid cells through an unclear mechanism. Flux through the glycolytic pathway in myeloid cells is inextricably linked to the availability of NAD⁺, which is maintained by NAD⁺ salvage and lactate metabolism. Using lung tissue from tuberculosis (TB) patients and myeloid deficient LDHA (*Ldha^LysM−/−*) mice, we demonstrate that glycolysis in myeloid cells is essential for protective immunity in TB. Glycolytic myeloid cells are essential for the early recruitment of multiple classes of immune cells and IFNγ-mediated protection. We identify NAD⁺ depletion as central to the glycolytic inhibition caused by *Mtb*. Lastly, we show that the NAD⁺ precursor nicotinamide exerts a host-dependent, antimycobacterial effect, and that nicotinamide prophylaxis and treatment reduce *Mtb* lung burden in mice. These findings provide insight into how *Mtb* alters host metabolism through perturbation of NAD(H) homeostasis and reprogramming of glycolysis, highlighting this pathway as a potential therapeutic target.

*Mycobacterium tuberculosis* (*Mtb*), the bacterium that causes tuberculosis (TB), remains a leading cause of death worldwide despite curative antibiotic therapy[1]. The success of *Mtb* as a pathogen is attributable to numerous virulence factors that modulate the host immune response to allow escape from host phagocytes, delay the onset of adaptive immunity, and cause the chronic inflammation and immunopathology characteristic of TB[2]. Several virulence factors have been well characterized since *Mtb* was first isolated in 1882; however, our understanding of the immunomodulatory strategy of this pathogen remains incomplete. A growing body of evidence indicates that host immunometabolism, the intrinsic link between metabolism and immune function, is an important component of TB pathogenesis[3–5]. In this regard, the metabolism of myeloid cells has emerged as a crucial determinant of TB outcomes, given that myeloid cells are the primary host reservoir for *Mtb*[6] and their inflammatory functions are directly responsible for controlling infection and the immunopathology that defines TB[7–10]. Within this subset, sulfur[11] and iron[12,13] metabolism as well as central carbon metabolism[14] play a role in host protection in TB.

Glycolysis is of particular interest[14–20] since it is the pathway in central carbon metabolism by which cells oxidize glucose to pyruvate, simultaneously converting ADP and NAD⁺ to ATP and NADH, respectively. While glycolysis is often thought of as coupled with mitochondrial respiration under aerobic conditions, myeloid cells maintain high levels of flux through this anaerobic pathway even in the presence of oxygen to drive inflammation across many systems, referred to as aerobic glycolysis or the Warburg effect[14,21]. Initial studies using

[1]Department of Microbiology, University of Alabama at Birmingham, Birmingham, AL, USA. [2]Africa Health Research Institute, University of KwaZulu Natal, Durban, South Africa. [3]Department of Laboratory Medicine and Pathology, Walter Sisulu University, Eastern Cape, South Africa. [4]Department of Medicine, Division of Nephrology, Nephrology Research and Training Center, University of Alabama at Birmingham, Birmingham, AL, USA. [5]Centers for AIDS Research and Free Radical Biology, University of Alabama at Birmingham, Birmingham, AL, USA. ✉e-mail: agarwal@uab.edu; asteyn@uab.edu

nonvirulent *Mtb* (i.e., killed or attenuated) suggest that *Mtb* infection prompts macrophages, a myeloid subset, to carry out aerobic glycolysis[22,23], but a growing body of literature indicates that live, virulent *Mtb* decreases the glycolytic capacity of macrophages following infection[16,19,20]. Other studies associate glycolytic flux in macrophages with bacillary control both in vitro and in vivo[15,17,18]. However, causally implicating glycolysis in these studies has been confounded by using experimental manipulations that lead to large-scale disruption of glycolysis and connected pathways, such as the use of 2-deoxy-D-glucose (2DG), an inhibitor of glucose uptake, and deletion of *Hif1A*, the gene encoding the highly conserved transcription factor, hypoxia inducible factor-1α (HIF1α). Consequences of inhibition with 2DG are a complete inhibition of glycolysis and dysregulation of the linked pentose phosphate pathway (PPP) and the TCA cycle, which uses the end-product of glycolysis, i.e., pyruvate, as a substrate. Similarly, HIF1α has been shown to broadly regulate the expression of enzymes involved in glycolysis[24] as well as transcriptional regulation, cytoskeletal organization, and other essential cellular processes[25]. Therefore, an experimental approach that selectively targets a distinct step in glycolysis during *Mtb* infection would provide compelling evidence for its role in the control of TB.

Glycolysis is an essential metabolic process and is regulated at multiple levels. One cell-intrinsic approach to regulating glycolytic flux would be to modulate the availability of $NAD^+$, the electron acceptor in glycolysis. In inflammatory myeloid cells, $NAD^+$ availability [i.e., NAD(H) homeostasis] is largely dependent on: (i) maintaining the overall abundance of NAD(H) via the $NAD^+$ salvage pathway, and (ii) regenerating $NAD^+$ from NADH via lactate fermentation. $NAD^+$ salvage regenerates $NAD^+$ from the precursor nicotinamide (NAM), which is also a product of $NAD^+$-consuming enzymes that are increased in inflammatory myeloid cells[26]. Lactate metabolism, on the other hand, couples the oxidation of NADH to $NAD^+$ with the reduction of pyruvate to lactate. The reversible process of lactate fermentation is catalyzed by lactate dehydrogenase (LDH), a tetramer composed of LDHA and LDHB subunits. When the tetramer is composed predominantly (or exclusively) of LDHA subunits, the subunit predominantly expressed in myeloid cells, it preferentially converts pyruvate to lactate, and NADH to $NAD^+$. However, LDHB subunits favor the opposite reaction. By reducing the availability of $NAD^+$, glycolytic flux is slowed but not completely disrupted, as is the case for direct disruption of glycolytic enzymes via deletion of HIF1α or administration of 2DG. Thus, the intrinsic link between NAD(H) homeostasis and glycolysis provides several targets for finer manipulation of host-cell glycolytic flux.

Given the unmet need for shorter, simpler, or more tolerable regimens for TB treatment, specific manipulation of host glycolysis represents a potential host-directed therapy (HDT). This approach would supplement existing TB antibiotic therapy by enhancing the microbicidal function and/or limiting inflammation caused by host immune cells, especially myeloid cells[27–29]. Thus, the explicit contribution of glycolytic capacity in myeloid cells to host protection against TB is unknown. We therefore tested the hypothesis that NAD(H)-mediated glycolytic flux in myeloid cells is essential for host protection in TB. We examined LDHA protein expression patterns in human TB lung tissue, employed a myeloid-specific *Ldha* knockout mouse (*Ldha*^LysM−/−) that exhibits reduced glycolytic capacity in myeloid cells[30,31], and performed a series of bioenergetic and metabolomic experiments in *Ldha*^LysM−/− macrophages. We applied these findings to in vitro and in vivo models of *Mtb* infection using a pharmacological inhibitor and nutritional supplementation. Our findings highlight the essential role for glycolytic myeloid cells in host protection and a mechanism by which *Mtb* disrupts glycolysis in these cells. Further, increasing the glycolytic capacity of myeloid cells in the context of TB could represent an innovative approach to the prevention and treatment of TB.

## Results

### LDHA expression within the spectrum of human tuberculosis

Given the complexity and diversity of lesions in pulmonary tuberculosis and the suboptimal representation of this spectrum within available animal models[32,33], we examined LDHA expression patterns in lung tissue resected from patients with TB to determine its clinical relevance across the histopathological spectrum of human pulmonary TB. In non-necrotizing granulomas (NNGs), we observed clusters of LDHA-positive cells with a lower nucleus:cytoplasm ratio, likely macrophages, and some lymphocytes, spread throughout a predominantly lymphocytic infiltrate (yellow arrows; Fig. 1A). In early necrotizing granulomas, we observed intense, homogenous LDHA staining of the necrotic core (Fig. 1B), representing LDHA released from necrotic immune cells. Heterogenous staining for LDHA is also present throughout areas of granulomatous inflammation and the developing necrotic core of more advanced necrotic lesions (Fig. 1C). Lastly, in established, necrotic granulomas, LDHA staining is limited to the granulation layer (Fig. 1D) and proximal regions of granulomatous inflammation. This stands in contrast to early necrotic granulomas, which exhibit robust staining of necrotic debris and illustrate a pattern consistent with a loss of staining with increased age of the lesion. Thus, LDHA expression, and consequently an increased capacity for lactate metabolism, is localized to regions of active inflammation within the human TB lung.

Further association of LDHA with immune cell function in the human TB lung can be made based on positive LDHA staining within giant cells (Fig. 1E, F; Fig. S1A), lymphoid aggregates (Fig. 1G), and alveolitis (Fig. 1H; Fig. S1B, C). The lack of LDHA staining in the immediate vicinity of these cells suggests that LDHA is linked to immune cell function as opposed to strictly a response to the surrounding microenvironment. LDHA is not expressed exclusively by immune cells in the human TB lung. We also observed positive LDHA staining in bronchial epithelial cells (Fig. 1I; Fig. S1D) and in the pulmonary vasculature (Fig. S1E, F). Of note, we observed the LDHA-positive neutrophils crossing the bronchial epithelium and LDHA-positive neutrophils and macrophages among the inflammatory debris within the airway, underscoring a potential role for LDHA in the function of these cells (Fig. 1I). The negative control for this staining includes an isotype control antibody (Fig. S2A–D), which showed immunonegative reactions, demonstrating the specificity of LDHA staining.

Taken together, histopathological appraisal of TB lesions provides new insight into the spatial distribution of LDHA within the human tuberculous lung. The distinct patterned responses within the spectrum of lesions were illustrated by myeloid, bronchial epithelial cells, and lymphocytes that stain positive for LDHA while engaging in distinctive immunological phenomena like granuloma formation and alveolitis. These data implicate LDHA as an important metabolic protein in the immune response in human TB lesions.

### Glycolytic capacity in myeloid cells protects mice against *Mtb* infection

Since NAD(H) regulates glycolysis at defined steps and the role of LDHA in TB pathogenesis is unknown, we hypothesized that NAD(H)-mediated glycolytic flux in myeloid cells protects the host against *Mtb* infection (Fig. 2A, B). We tested this hypothesis using *Ldha*^LysM−/− mice which lack LDHA in the myeloid compartment[31] (Fig. 2C, D). Myeloid cells in these mice exhibit reduced LDH function as evidenced by a decreased ability to regenerate $NAD^+$ from NADH in the presence of pyruvate (Fig. 2E), which consequently reduces their glycolytic capacity (Fig. 2F, G). We infected *Ldha*^LysM−/− and *Ldha*^fl/fl (WT) mice with a low-dose (~30 colony forming units [CFU]) of *Mtb* H37Rv to assess their survival in a chronic model of TB. *Ldha*^LysM−/− mice were more susceptible to *Mtb* infection with significantly reduced survival (Fig. 2H). We assessed the burden of *Mtb* and the pathology in the lungs and spleen

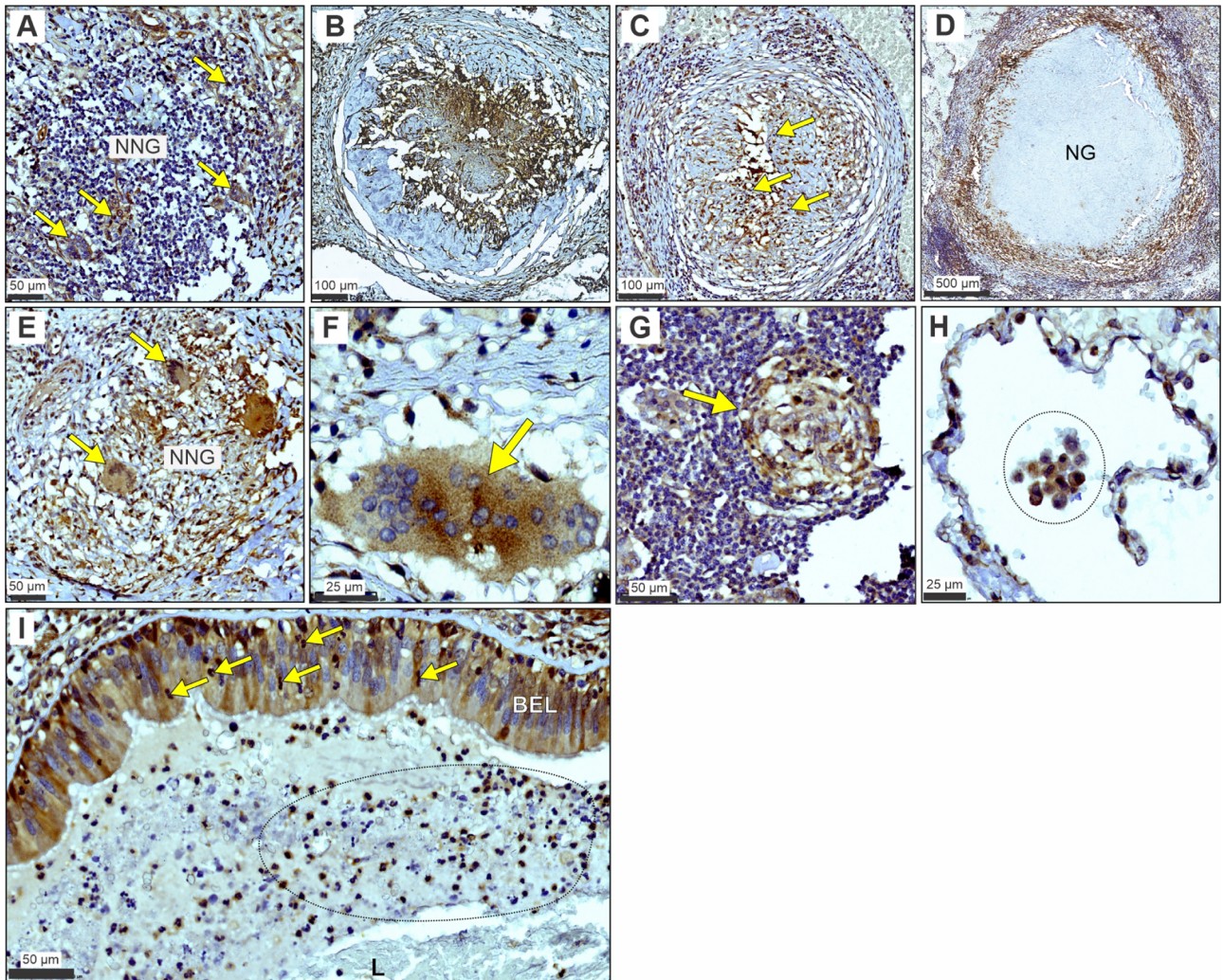

**Fig. 1 | LDHA immunostaining across the histopathological spectrum of human TB. A** Focal staining of LDHA (yellow arrows) within a non-necrotic granuloma (NNG). **B** Homogenous staining of LDHA within the necrotic center of an early necrotic granuloma. **C** Heterogeneous staining of LDHA within the necrotic center of a more developed necrotic lesion. Yellow arrows indicate pyknotic nuclei within the necrotic center. **D** Staining of LDHA in the granulation layer and surrounding granulomatous inflammation region of a necrotic granuloma (NG). **E** Medium- and (**F**) high-magnification images of giant cells immunostained for LDHA (yellow arrows) in the context of (**E**) a non-necrotizing granuloma (NNG) and (**F**) necrotizing granuloma. **G** LDHA staining of a lymphoid aggregate (yellow arrow) in a region proximal to a NG. **H** High-magnification image of LDHA staining in leukocytes (circled region) in an alveolus of a human TB patient. **I** LDHA staining in neutrophils (yellow arrows) crossing the positively-stained bronchial epithelial layer (BEL) in the context of inflammatory and necrotic debris (marked oval region) within the airway lumen (L). Tissue staining and staining controls obtained from three TB patients (Table S1) were performed at least three times independently post optimization.

of similarly low-dose infected mice at 4 weeks post infection (wpi), 10 wpi, and 30 wpi, which correspond to the early, middle, and late stages of the chronic TB model, respectively. At 4 wpi and 10 wpi, the *Mtb* burden in the lungs of *Ldha*[LysM−/−] mice was significantly increased compared to WT mice. However, there was no difference in lung burden at 30 wpi (Fig. 2I). The *Mtb* burden in the spleens of *Ldha*[LysM−/−] mice was marginally increased at 4 wpi compared to WT, but no significant differences in *Mtb* burden were found at any time point (Fig. S3A).

Consistent with overall survival, qualitative assessment of the gross pathology (Fig. S3B) and histopathology (Fig. 2J) of the lungs revealed worse disease in *Ldha*[LysM−/−] mice at 30 wpi. Interestingly, while lesions were well organized with a very high cell density at 30 wpi, inflammation was more diffuse at 4 wpi (Fig. 2J). To accurately quantify this diffuse pathology, we took an unsupervised approach using the open-source application, QuPath[34], to segment nuclei across each tissue section and determine the local density of cells for each nucleus. We found that WT mice initially mounted a robust inflammatory response that resolved by 30 wpi (Fig. S3C), consistent with a more protective immune response to *Mtb* infection. We applied this approach to lungs from *Mtb*-infected *Ldha*[LysM−/−] mice and found a striking absence of early inflammation in *Ldha*[LysM−/−] mice compared to WT controls (Fig. 2K, L; Fig. S4). This deficit persisted through 10 wpi, and, while WT mice appeared to undergo significant resolution of inflammation at 30 wpi, inflammation in the lungs of *Ldha*[LysM−/−] mice progressed through this time point (Fig. 2K, L; Fig. S4).

Altogether, we found that *Ldha*[LysM−/−] mice are more susceptible to TB, maintain a higher *Mtb* burden in the lungs, and exhibit a dysfunctional inflammatory response to *Mtb* infection. This suggests that LDHA is necessary for protection against TB and that glycolytic flux in myeloid cells is essential for the control of *Mtb* infection and disease.

## Glycolytic myeloid cells are essential for protective immunity in TB

We observed reduced inflammation in the lungs of *Ldha*[LysM−/−] mice at 4 wpi and increased inflammation at 30 wpi. To determine whether the

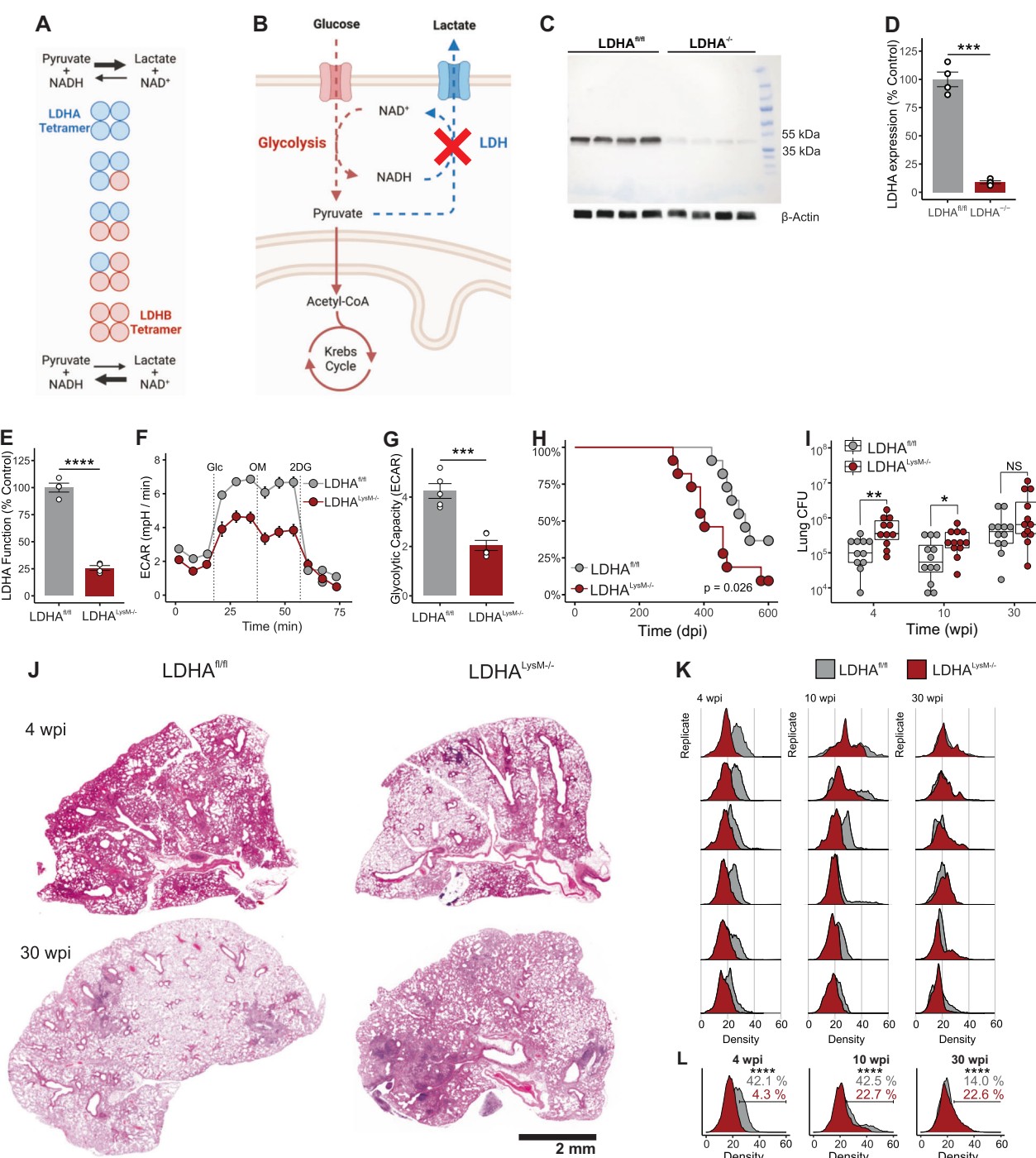

**Fig. 2 | *Ldha^{LysM−/−}* mice are more susceptible to TB. A** Depiction of the favored conversion of pyruvate to lactate by LDH composed predominantly of LDHA subunits compared to the reverse reaction by LDH composed predominantly of LDHB subunits. **B** Depiction of the glycolytic defect imposed by LDHA deletion. **C** Immunoblot showing relative LDHA protein levels in LDHA^{−/−} and LDHA^{fl/fl} BMDMs. **D** LDHA protein levels and (**E**) LDH function in protein lysates normalized to the mean values for LDHA^{fl/fl} BMDMs. Symbols represent biological replicates from an independent experiment (n = 4/group). **F** ECAR of LDHA^{fl/fl} and LDHA^{−/−} BMDMs. Dashed lines indicate the injection of glucose, oligomycin, and 2-DG. Symbols and error bars represent mean ± SEM of 5 technical replicates. **G** Columns, error bars, and symbols represent the mean, SEM, and individual values for glycolytic capacity determined from data in (**F**). **H** Kaplan-Meier plot of mice infected with ~30 CFU of *Mtb* (n = 11/group). **I** Box plot of *Mtb* burden in mouse lungs, where the center, upper and lower hinges, and upper and lower whiskers indicate the 2nd quartile, 3rd and 1st quartile, and ±1.5 times the interquartile range (3rd quartile −

1st quartile), respectively. Symbols represent biological replicates pooled from two independent experiments (n ≥ 10/group). **J** H&E staining of representative lung sections from *Mtb*-infected mice (n = 6 per group from one experiment) at indicated times post infection. **K** Normalized histograms representing cellular density around each nucleus in a tissue section. Within each timepoint, biological replicates were ranked by mean density and plotted with the replicate of the corresponding rank from the other genotype. **L** Cumulative histogram of all replicates compared by genotype at each time point. Percentages indicate the proportion of cells in each group above a threshold density of 25. Data in (**D**) and (**E**) are shown as individual values with the group mean ± SEM. Statistical significance was determined by two-sided, two-sample *t*-test not assuming equal variance (**D, E, G**); log-rank test (**H**); two-sided, two-sample Wilcoxon rank-sum test (**I**); and two-sample *z*-test (**L**). *p < 0.05, **p < 0.01, ***p < 0.001, ****p < 0.0001; exact *p*-values for each comparison are listed in Supplementary Data 1. Source data are provided as a Source Data file.

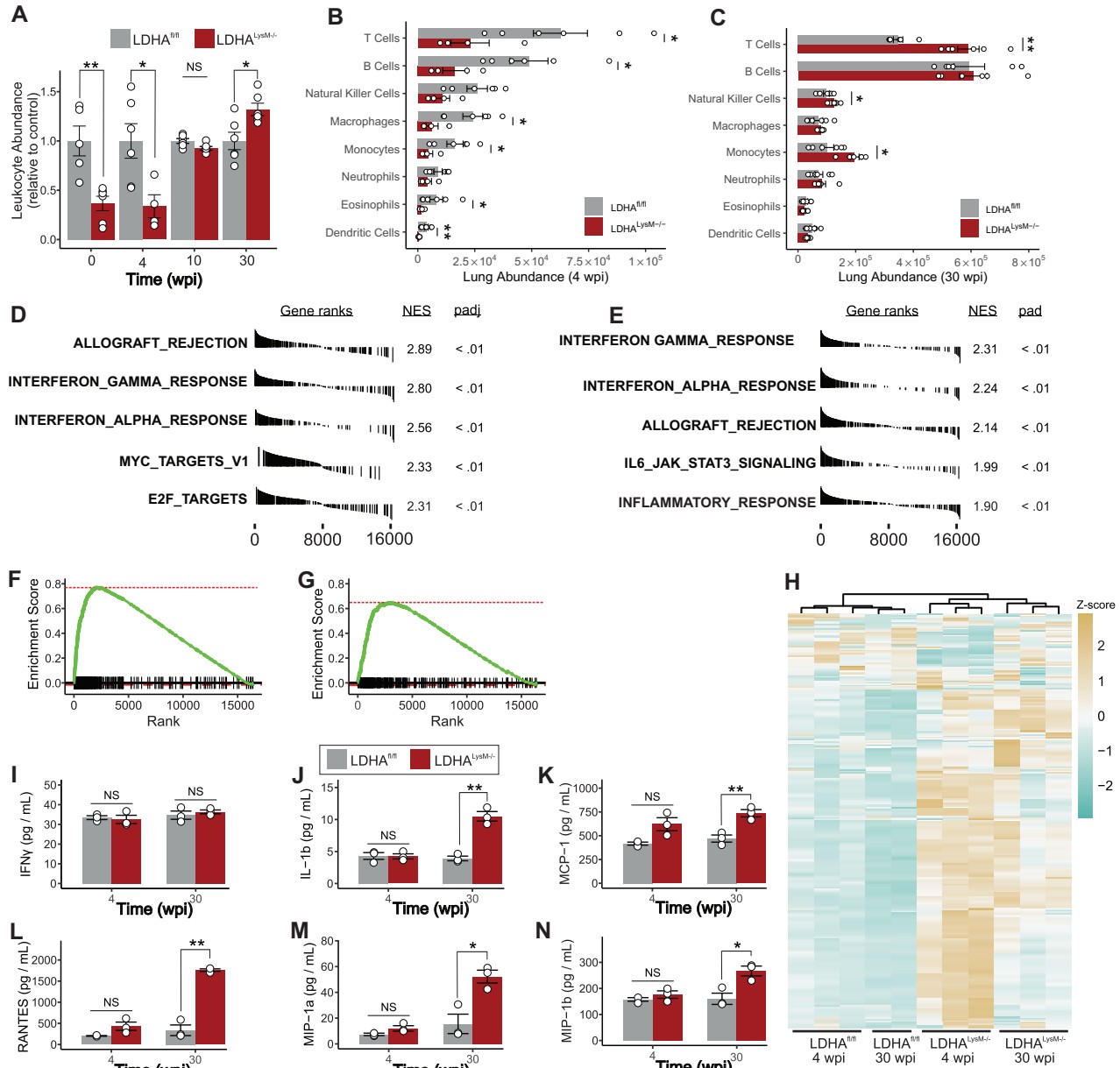

**Fig. 3 | *Ldha^LysM−/−* mice exhibit a dysregulated immune response to *Mtb* infection. A** Mean abundance of leukocytes within the lungs of *Mtb*-infected mice at times post infection (*n* ≥ 4/group). **B**, **C** Absolute number of immune cells within indicated populations in the lungs of *Mtb*-infected mice at (**B**) 4 wpi and (**C**) 30 wpi (*n* = 6/group). **D**–**G** The most enriched hallmark gene sets (**D**, **E**) and enrichment plots (**F**, **G**) for the transcriptional response to IFNγ stimulation in the lungs of infected *LDHA^LysM−/−* mice compared to infected *LDHA^fl/fl* mice at 4 wpi (**D**, **F**) and 30 wpi (**E**, **G**). **H** Heatmap representing the expression of the genes comprising the hallmark response to IFNγ gene set. Color-scale values represent the *z*-score for the expression in a given row and the dendrogram above the columns shows hierarchical clustering based on the expression of these genes. **I**–**N** Cytokine levels in the lungs of *Mtb*-infected mice (*n* = 3/group). Data are presented as individual values with the group mean ± SEM for panels (**A**–**C**) and (**I**–**N**). Statistical significance was determined by the two-sided, two-sample Wilcoxon rank-sum test (**A**–**C**) and two-sided, two-sample *t*-test not assuming equal variance. *$p < 0.05$, **$p < 0.01$; exact *p*-values for each comparison are listed in Supplementary Data 1. Source data are provided as a Source Data file.

dysfunctional immune response to *Mtb* infection was generalized across all leukocytes or unique to a particular subset or lineage, we used multiparameter flow cytometry to phenotype and quantify the immune cells trafficking to the lungs of *Ldha^LysM−/−* mice in our chronic model of TB (Fig. S5A)[35]. The relative abundance of live, CD45+ cells (leukocytes) showed a trend consistent with our histology data, with *Ldha^LysM−/−* mice exhibiting decreased abundance of leukocytes at 4 wpi infection and increased abundance at 30 wpi, relative to WT controls (Fig. 3A). The differences in leukocyte abundance at 0 wpi and 4 wpi arose from differences in abundance of several immune cell types (Figs. 3B, S4B), but was limited to neutrophils and T cells at 10 wpi

(Fig. S5C) and T cells, natural killer cells, and monocytes at 30 wpi (Fig. 3C). Of note, cell types belonging to both lymphoid and myeloid lineages were differentially abundant at all timepoints, indicating the origin of these differences lies in the recruitment of immune cells as opposed to differences in the trafficking or the viability of myeloid cells due to their glycolytic deficiency.

To better understand functional aspects of the immune response in the lungs of *Ldha^LysM−/−* mice, we performed RNA sequencing (RNA-Seq) in the lungs of these mice at 4 wpi and 30 wpi. To assess global differences in lung transcriptomes, we performed principal component analysis, which separates replicates by group across the first two

components (Fig. S5D). To identify the biological processes that best distinguished these groups, we analyzed differentially expressed genes using gene set enrichment analysis (GSEA) to identify crucial biological processes for which transcripts are enriched or deficient in *Ldha*<sup>LysM-/-</sup> mice[36]. Seemingly contrary to our evidence of a reduced immune response at 4 wpi, transcripts associated with inflammatory processes were among the most enriched transcripts in *Ldha*<sup>LysM-/-</sup> mice at 4 wpi and 30 wpi among the 50 hallmark gene sets curated for GSEA (Fig. 3D, E). In particular, the response to IFNγ was the most enriched gene set at both time points (Fig. 3D–G). This relationship was further underscored by the ability of hierarchical clustering to separate the groups of samples based on their expression of the genes in this gene set (Fig. 3H). The robust IFNγ gene signature in more susceptible mice with a blunted immune response is particularly intriguing since IFNγ is an indispensable antimycobacterial cytokine widely considered to be protective in TB[37,38].

To determine whether the enrichment of mRNA corresponded to increased levels of cytokines and chemokines, we performed a multiplexed, fluorescence-based assay to quantify the expression of cytokines and chemokines in lung tissue from the same mice for which sequencing data was available. Despite an increase in transcription, we observed similar levels of nearly all cytokines and chemokines between WT and *Ldha*<sup>LysM-/-</sup> mice at 4 wpi, including IFNγ (Fig. 3I–N; Fig. S5E–G). However, at 30 wpi levels of several key cytokines and chemokines were increased in *Ldha*<sup>LysM-/-</sup> animals compared to WT controls, consistent with other signs of chronic inflammation (Fig. 3J–N; Fig. S5E–G). Consistent with our flow cytometry data, we observed increased concentrations of chemokines associated with the recruitment of monocytes (MCP-1, KC), T cells (RANTES), or both (MIP-1a, MIP-1b).

Taken together, these results indicate glycolytic myeloid cells are important for the early recruitment of both lymphocytes and other myeloid cells in response to *Mtb* infection. Furthermore, given the discrepancy between having a robust IFNγ transcriptional signature and increased bacterial burden in the lungs of *Ldha*<sup>LysM-/-</sup> mice, our data complement previous studies[15,18,22] by suggesting that glycolytic myeloid cells are essential mediators of the protective effects of IFNγ in vivo.

**Macrophages require LDHA for their metabolic response to IFNγ**

Based on our RNA-Seq analysis showing increased IFNγ signaling in the context of poorly controlled *Mtb* infection in *Ldha*<sup>LysM-/-</sup> mice, we next sought to understand the role of lactate metabolism in the myeloid response to IFNγ. We hypothesized that LDHA-deficient macrophages are unable to respond metabolically to IFNγ. Therefore, we performed combined glycolysis/mitochondrial stress tests on bone marrow-derived macrophages (BMDMs)[39]. Because this technique determines the non-glycolytic acidification caused by cells based on the extracellular acidification rate (ECAR) prior to the injection of glucose rather than after the injection of 2DG, it is important to validate that these values are the same in a standard glycolysis stress test. Indeed, we found no difference in these two approaches in a standard glycolysis stress test (Fig. S6A, B). Consistent with our hypothesis, we observed only a modest difference in glycolytic capacity between LDHA<sup>-/-</sup> and WT BMDMs without IFNγ (Fig. 4A). However, upon stimulation with IFNγ, the increase in glycolytic capacity for LDHA<sup>-/-</sup> BMDMs was threefold lower than in WT BMDMs (Fig. 4A, B; Fig. S6C–E). Furthermore, while mitochondrial respiration in LDHA<sup>-/-</sup> BMDMs was unchanged by stimulation with IFNγ (Fig. 4C, D), these cells were more reliant on OXPHOS at baseline (Fig. S6G–J) and had reduced spare respiratory capacity (SRC) (Fig. S6K). Last, parameters not associated directly with energy production were largely unaffected by LDH inhibition (Fig. S6L–N). Taken together, these data demonstrate that LDHA<sup>-/-</sup> macrophages exhibit no bioenergetic reserve (Fig. 4E, F), hampering their metabolic response to IFNγ.

In our initial combined stress tests, injection of FCCP and pyruvate elicited a marked increase in ECAR in LDHA<sup>-/-</sup> BMDMs (Fig. S6F). We posited that supplementing LDHA<sup>-/-</sup> BMDMs with pyruvate would increase lactate fermentation catalyzed by LDH composed of LDHB subunits, with a concomitant increase in limiting NAD<sup>+</sup>. We found that while pyruvate supplementation did little to impact basal glycolytic flux (Fig. S6D), it rescued the deficit in the glycolytic response to IFNγ (Fig. 4A, B) and the deficit in glycolytic reserve in LDHA<sup>-/-</sup> BMDMs (Fig. S6E), thereby reducing their deficit in glycolytic capacity (Fig. S6C). Importantly, this rescue was attenuated by the selective LDH inhibitor GSK 2837808A, indicating the effect is mediated by residual LDH function (i.e., LDHB) in LDHA<sup>-/-</sup> BMDMs (Fig. 4A, B; Fig. S6C–E). Finally, this pattern was also observed with regard to bacterial control. Specifically, LDH-deficiency in BMDMs had no impact on bacterial replication at baseline (Fig. 4G). However, upon stimulation with IFNγ, LDHA<sup>-/-</sup> macrophages exhibited a marginally increased bacterial burden. As with the change in glycolytic capacity in response to IFNγ stimulation, pyruvate equalized the difference in bacterial burden, while GSK 2837808A prevented the effect of pyruvate. In this experiment we also observed the previously described increase in bacterial burden following pyruvate supplementation[18]. The abrogation of this effect with LDH inhibition suggests GSK 2837808A may also inhibit bacterial LDH.

To further characterize the glycolytic metabolism of LDHA<sup>-/-</sup> BMDMs, we assessed the NAD<sup>+</sup>:NADH ratio and glucose uptake in these cells following stimulation with IFNγ and pyruvate supplementation. LDHA<sup>-/-</sup> BMDMs exhibited a decreased NAD<sup>+</sup>:NADH ratio regardless of treatment condition (Fig. 4H), suggesting either that LDHA<sup>-/-</sup> BMDMs have adapted to maintaining a decreased ratio or that, while pyruvate may help to increase the flux of NAD<sup>+</sup> regeneration, this increase still does not meet the demand for NAD<sup>+</sup> in LDHA-deficient macrophages. Interestingly, however, glucose uptake was no different between LDHA<sup>-/-</sup> and WT BMDMs (Fig. S5O).

Considering the comparable glucose uptake but diminished metabolic flux through glycolysis, we hypothesize that IFNγ-stimulated LDHA<sup>-/-</sup> BMDMs would accumulate metabolic intermediates within glycolysis and the reversible reactions of the PPP. Therefore, we performed targeted metabolomics with uniformly labeled $^{13}C_6$ glucose, which revealed a cluster of metabolites (F16BP, PGA, PEP, E4P, and G3P) with significantly increased abundance in LDHA<sup>-/-</sup> BMDMs which falls between the rate-limiting steps in upper and lower glycolysis and the irreversible oxidative PPP (Fig. 4I). While abundances alone do not reflect flux, analysis of the distribution of mass isotopologues revealed increased incorporation of $^{13}C$ into the intermediates of upper glycolysis and the PPP in LDHA<sup>-/-</sup> BMDMs, but comparable incorporation into the intermediates of lower glycolysis (Fig. 4K). This demonstrates that carbon flux is increased through the initial reactions of glycolysis in LDHA-deficient macrophages, but more carbon is diverted to the PPP, consistent with limited flux through the NAD<sup>+</sup>-dependent reaction catalyzed by glyceraldehyde phosphate dehydrogenase. As with our bioenergetics analysis, pyruvate supplementation abrogated differences in both accumulation and flux (Fig. 4J, K; Fig. S6P). Interestingly, while the observed cluster of glycolytic and PPP metabolites was no longer present in pyruvate-supplemented BMDMs, metabolites in a new cluster were maintained at reduced abundance in LDHA<sup>-/-</sup> BMDMs (Fig. 4J). This new cluster is comprised of important TCA cycle intermediates, such as citrate, as well as amino acids (e.g., Val, Asn, Leu, Asp, Gln, Ile) that are carbon sinks for TCA cycle intermediates. These findings are consistent with excess pyruvate being utilized by WT BMDMs as a carbon source for the TCA cycle, whereas LDHA<sup>-/-</sup> BMDMs primarily rely on pyruvate as an electron sink to regenerate NAD<sup>+</sup>.

In summary, these data suggest that LDH-mediated NAD<sup>+</sup> regeneration is essential for the metabolic response to IFNγ. Of note, we also found that pyruvate supplementation can have the counterintuitive effect of facilitating glycolytic flux in the setting of reduced LDH

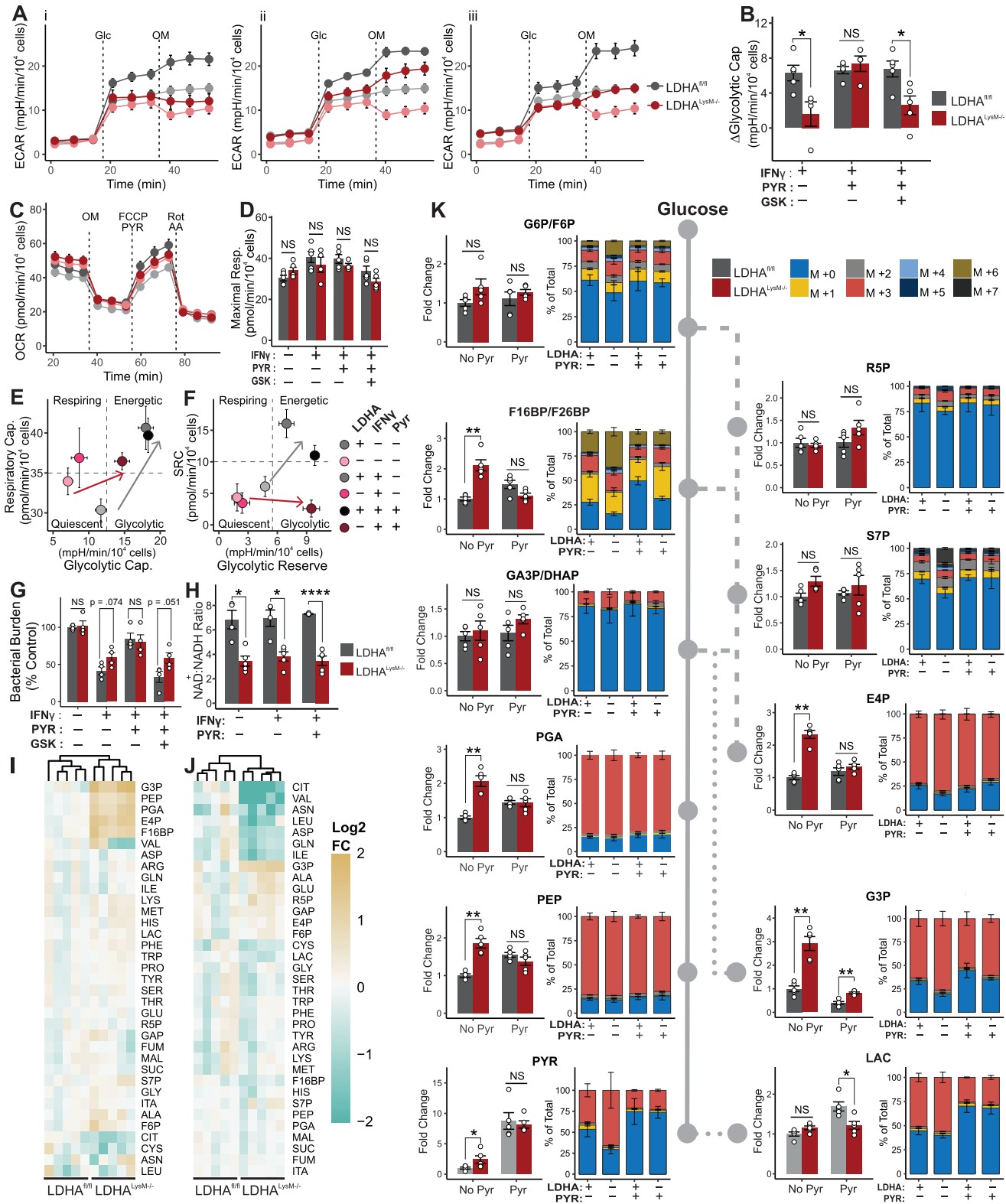

enzymatic capacity, opening the door for a new approach to the enhancement of glycolysis under appropriate conditions.

### *Mtb* disrupts NAD(H) homeostasis in macrophages

Our data show that glycolysis in myeloid cells, which depends strictly on NAD⁺ availability, is essential for protective immunity mediated by IFNγ. Given that virulent *Mtb* also reduces glycolytic flux in macrophages relative to killed or attenuated *Mtb*[16,19,20], we next hypothesized

that, like *LDHA* deletion, virulent *Mtb* blunts the metabolic response to IFNγ in infected macrophages. Consistent with our hypothesis, *Mtb* infection increased the glycolytic capacity of WT BMDMs relative to no stimulus; however, these infected cells were unable to increase their glycolytic capacity in response to IFNγ stimulation compared to uninfected BMDMs (Fig. 5A, B). Unlike LDHA⁻/⁻ BMDMs, however, the blunted response to IFNγ induced by *Mtb* infection could not be rescued by pyruvate supplementation (Fig. S7A, B), suggesting a

**Fig. 4 | LDHA$^{-/-}$ BMDMs exhibit pyruvate-reversible deficiency in their metabolic response to IFNγ. A** ECAR of LDHA$^{-/-}$ (red) and WT (gray) BMDMs treated with (i) IFNγ (10 ng/mL); (ii) IFNγ and 1 mM pyruvate; and (iii) IFNγ, pyruvate, and 5 μM of the LDH inhibitor GSK 2837808A (darker lines and symbols) relative to untreated BMDMs (lighter lines and symbols). Dashed lines indicate injection of glucose (Glc) followed by oligomycin (OM). Symbols and error bars represent mean ± SEM of 5 biological replicates. **B** Change in glycolytic capacity of treated BMDMs compared to unstimulated BMDMs (*n* = 5/group) determined from the profiles in (**A**). **C** OCR of BMDMs treated with 10 ng/mL IFNγ (darker lines and symbols) or no stimulus (lighter lines and symbols). Dashed lines indicate injection of OM, FCCP/Pyruvate, or Rotenone/Antimycin A. Symbols and error bars represent mean ± SEM of 5 biological replicates. **D** Maximal respiration of BMDMs treated as indicated (*n* = 5/group). **E**, **F** Phenograms for the indicated parameters extracted from the XF profiles of BMDMs under the indicated conditions. Symbols and error bars represent the mean ± SEM for 5 biological replicates. **G** Bacterial burden and (**H**) NAD$^+$:NADH ratio of BMDMs treated as indicated (*n* = 4/group). **I**, **J** Heatmaps representing the abundance of the metabolites comprising key pathways of central carbon metabolism in BMDMs treated with (**I**) 10 ng/mL IFNγ or (**J**) 10 ng/mL IFNγ and 1 mM pyruvate. Color-scale values represent the Log$_2$ fold-change relative to the mean value for WT BMDMs. Upper dendrogram represents hierarchical clustering of replicates based on metabolite abundance. **K** Abundance (left) and isotopologue distribution (right) of the indicated metabolite and its position within glycolysis (solid line), the PPP (dashed line), and related pathways (dotted lines). Fold change (FC) (left) relative to the mean of WT BMDMs stimulated with IFNγ. Stacked graphs (right) correspond to 100% of the total abundance of each metabolite, where the proportion of each mass isotopologue for each metabolite is indicated by color (*n* = 4/group). Data are presented as the group mean ± SEM for panels (**B**, **D**, **G**, **H**, and **K**) while points represent the individual values for biological (**B**, **D**, **H**, and **K**) or technical replicates (**G**). Statistical significance was determined by two-sided, two-sample *t*-test without assuming equal variance (**B**, **D**, **G**, **H**) or by two-sided, two-sample Wilcoxon rank-sum test (**K**). *$p < 0.05$, **$p < 0.01$, ****$p < 0.0001$; exact *p*-values for each comparison are listed in Supplementary Data 1. Source data are provided as a Source Data file.

mechanism distinct from reduced LDH expression. We therefore performed targeted metabolomics to characterize the central carbon metabolism in *Mtb*-infected BMDMs with or without IFNγ. We found *Mtb* infection decreased the abundance of glycolytic and PPP intermediates in BMDMs, except for pyruvate, which was significantly increased compared to uninfected macrophages with or without IFNγ-stimulation (Fig. 5C). In addition, analysis of TCA cycle intermediates revealed a comparable increase in the pool of (iso)citrate in *Mtb*-infected BMDMs. These data were complimented by analysis of the carbon isotopologue distribution. We observed increased labelling of G6P/F6P and PPP intermediates, such as sedoheptulose-7-phosphate (S7P) and erythrose-4-phosphate (E4P). This was in contrast to decreased labelling of F16BP/F26BP, the intermediates of lower glycolysis, including phosphoglyceraldehyde (PGA), phosphoenolpyruvate (PEP), and pyruvate (PYR), and the TCA cycle—with the exception of citrate (CIT; Fig. S7C, D). Together, these data strongly suggest a mechanism whereby the capacity of macrophages to funnel pyruvate into lactate metabolism is limited, leading to an accumulation of pyruvate, and subsequently, citrate, which then allosterically inhibits the rate-limiting enzyme of glycolysis, phosphofructokinase, further reducing glycolytic flux[40].

This mechanism suggests that while *Mtb*-infected macrophages likely have the enzymatic capacity to metabolize excess pyruvate to lactate, they are unable to do so. We then considered the contribution of NADH, given that it is the only other factor required for this reaction. *Mtb* was shown to reduce NAD(H) levels in infected macrophages[41,42], and our own bioenergetic and metabolomic analyses are consistent with the metabolic profile of inflammatory macrophages following inhibition of the NAD$^+$ salvage pathway (Fig. 5D)[26]. Based on these considerations, we hypothesize that glycolysis is inhibited in *Mtb*-infected BMDMs due primarily to a reduction in NAD(H) availability. Consistent with our hypothesis, while IFNγ stimulation increases the NAD(H) pool in macrophages, *Mtb* infection reduces the abundance of NAD(H) whether or not IFNγ is present (Fig. 5E). This decrease could be partially rescued by exposing macrophages to nicotinamide (NAM), which can be converted to NAD(H) via the NAD$^+$ salvage pathway. This rescue, as well as the residual NAD(H) pool in these cells, were eliminated by FK866, an inhibitor of nicotinamide phosphoribosyltransferase (NAMPT) the rate-limiting enzyme in NAD$^+$ salvage (Fig. 5E). To determine if the decreased glycolytic capacity in these cells resulted from reduced NAD(H) levels, we performed bioenergetic analysis on *Mtb*-infected BMDMs supplemented with NAM. Addition of NAM restored the glycolytic capacity of *Mtb*-infected BMDMs and FK866 blocked this effect (Fig. 5F, G). Consistent with FK866-mediated reduction in NAD(H) levels, treatment with FK866 further reduced glycolytic capacity below that of *Mtb*-infected, IFNγ-stimulated BMDMs. This suggests that NAD$^+$ salvage is partially responsible for

the maintenance of glycolytic capacity following infection with *Mtb*, even in the absence of NAM.

Lastly, we explored whether a similar process happens in human monocyte-derived macrophages (hMDMs). We found that hMDMs infected with *Mtb* exhibited an MOI-dependent decrease in NAD(H) abundance, and, likewise, this decrease could be prevented by treatment with NAM (Fig. S7E). Under the same conditions, hMDMs infected with *Mtb* exhibited an MOI-dependent decrease in glycolytic capacity. Interestingly, treatment with NAM had two distinct effects on glycolytic capacity in *Mtb*-infected hMDMs. First, it prevented an increase in glycolysis at a low MOI, and second, it prevented the MOI-dependent decrease in glycolytic capacity (Fig. S7F).

In summary, these data demonstrate that *Mtb* reduces NAD(H) availability, and that NAM maintains the glycolytic capacity of *Mtb*-infected macrophages by restoring NAD(H) levels through its conversion to NAD(H) via the NAD$^+$ salvage pathway. Furthermore, these findings underscore the importance of mechanistic insight into the pathogenesis of *Mtb*, as pyruvate supplementation, which restored the glycolytic capacity of *LDHA$^{-/-}$* macrophages (Fig. 4B), was ineffective at restoring the glycolytic capacity of WT macrophages infected with *Mtb* (Fig. S7A, B).

## Nicotinamide is an effective treatment for TB

NAM is known to be an effective treatment for TB[43–45]. It exerts a pH-dependent direct antimycobacterial effect dependent on bacterial pyrazinamidase (PncA)[46,47], and it has recently been shown to exert a PncA-independent host-directed antimycobacterial effect, as well[48]. Based on our observations that NAM increases the glycolytic capacity of *Mtb*-infected BMDMs and that glycolysis in myeloid cells promotes bacterial control in vivo, we hypothesize that NAM acts as a host-directed therapy by enhancing glycolysis in *Mtb*-infected macrophages through its conversion to NAD(H) via the NAD$^+$ salvage pathway. Therefore, we infected BMDMs with luciferase-expressing *Mtb* and measured luminescence at 48 hpi across a range of NAM conditions. We observed that NAM exerts a dose-dependent reduction in bacterial luminescence signal in BMDMs (Fig. 6A). Consistent with previous work[48], exposure to NAM only modestly reduced *Mtb* growth in 7H9 broth. We also tested the ability of NAM to reduce the bacterial burden in hMDMs. As with NADH abundance and glycolytic capacity, we observed an MOI dependent effect of NAM on its ability to reduce bacterial burden relative to untreated macrophages at the indicated MOI (Fig. S7G). Together, these data in 7H9 broth, BMDMs, and hMDMs suggest NAM exerts a host-dependent antimycobacterial effect.

We next measured *Mtb*-burden in infected BMDMs in the presence of NAM with the addition of either FK866 or 2DG. In both cases, we observed that the effect of NAM, but not the effect of the pathogen-

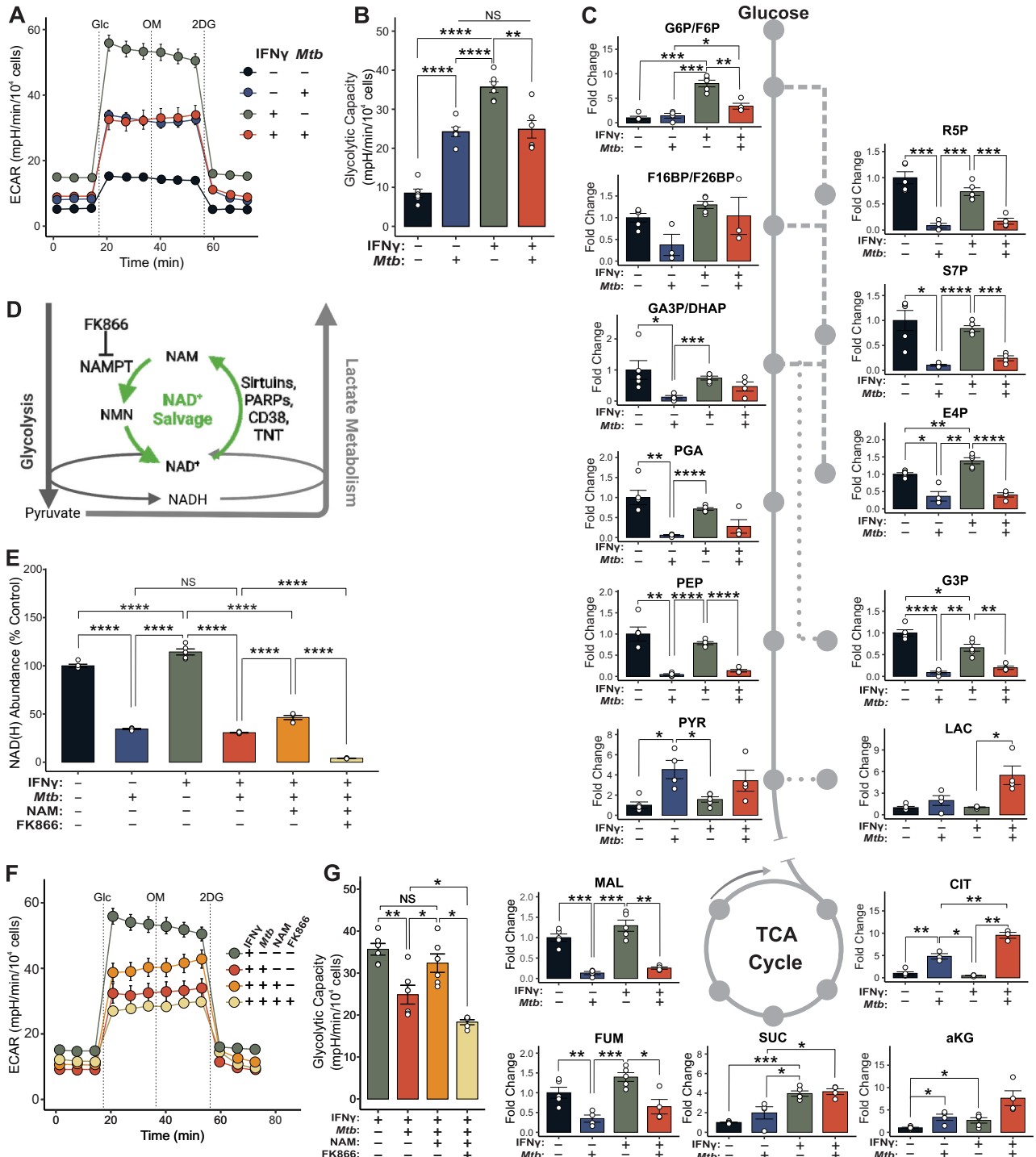

**Fig. 5 | *Mtb* disrupts glycolysis by reducing NAD(H) abundance in infected macrophages. A** ECAR profiles for WT BMDMs exposed to combinations of IFNγ (10 ng/mL) and *Mtb* (MOI 5:1) for 18 h. Dashed lines indicate injection of glucose (Glc), oligomycin (OM), or 2DG, respectively. Symbols and error bars represent mean ± SEM of 6 biological replicates. **B** Glycolytic capacity determined from the profiles in (**A**) (*n* = 6/group). **C** Abundance of metabolites and position within glycolysis (solid line), the PPP (dashed line), the TCA cycle (solid circle), and related pathways (dotted lines) relative to the mean of uninfected, unstimulated LDHAᶠˡ/ᶠˡ BMDMs (*n* = 4/group). Values for IFNγ-treated BMDMs are repeated from Fig. 4K. **D** The NAD⁺ salvage pathway (green), key enzymes, and the NAMPT inhibitor, FK866, and its relation to glycolysis and lactate metabolism. **E** Total abundance of NAD(H) within BMDMs under the indicated conditions (*n* = 5/group). **F** ECAR

profiles for BMDMs exposed to combinations of IFNγ (10 ng/mL), *Mtb* (MOI 5:1), NAM (1 mM) and FK866 (200 nM) for 18 h. Dashed lines represent injections of Glc, OM, or 2DG. Symbols and error bars represent mean ± SEM of 6 biological replicates. Values for IFNγ and IFNγ + *Mtb* are from (**A**). **G** Glycolytic capacity determined from the profiles in panel (**F**) (*n* = 6/group). Values for IFNγ and IFNγ + *Mtb* groups are from (**B**). Data are presented as individual values for biological replicates with the group mean ± SEM for panels (**B**, **C**, and **G**) and as individual values for technical replicates with the group mean ± SEM for panel (**E**). Statistical significance was determined by two-sided, two-sample t-test without assuming equal variance (**B**, **C**, **E**, **G**). *$p < 0.05$, **$p < 0.01$, ***$p < 0.001$, ****$p < 0.0001$; exact *p*-values for each comparison are listed in Supplementary Data 1. Source data are provided as a Source Data file.

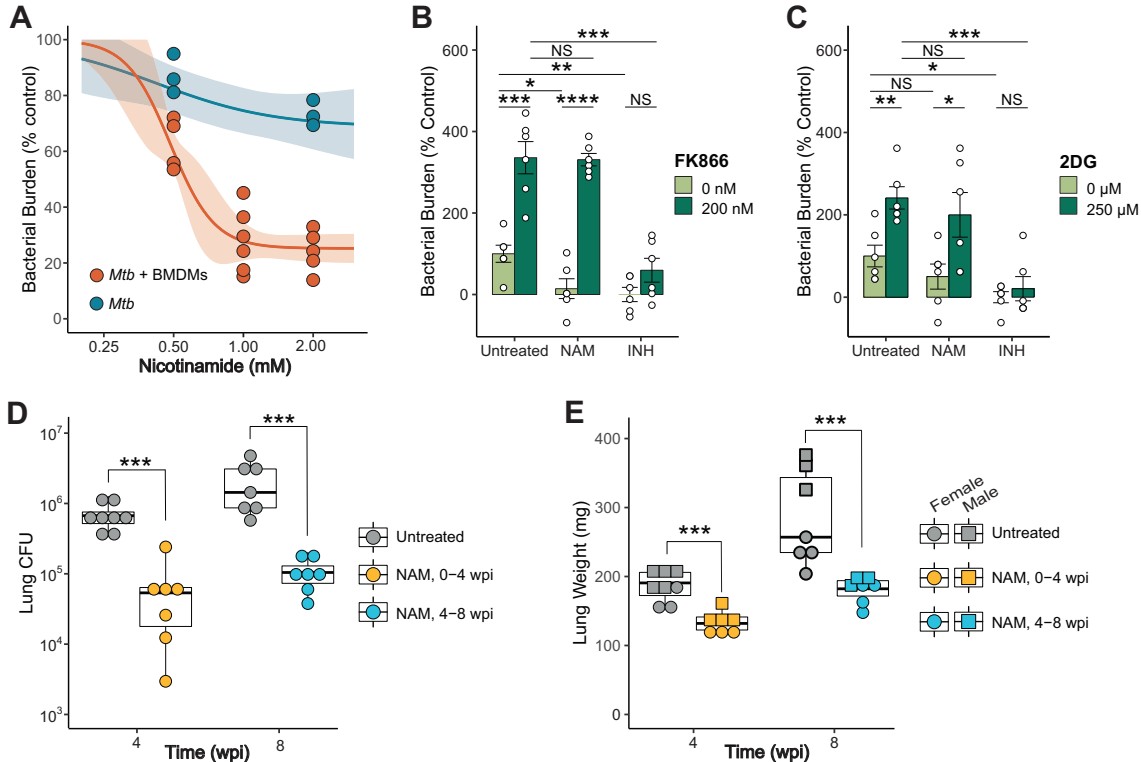

**Fig. 6 | Nicotinamide is an effective therapy for TB. A** Dose-response curves for the antimycobacterial effect of NAM on *Mtb* in 7H9 broth (blue) or *Mtb*-infected BMDMs (orange), calculated as described in Methods. The shaded region around each curve represents the 95% confidence interval, while dots represent biological replicates tested across indicated concentrations. **B, C** Bacterial burden in BMDMs infected with *Mtb* expressing a luciferase reporter treated with NAM and FK866 (**B**) or NAM and 2-DG (**C**) (*n* = 6/group). Isoniazid (INH) was included as a direct-acting antimycobacterial control. Data are shown as a percentage of the mean value of the untreated, uninfected group. **D, E** Box plots representing the lung burden of *Mtb* (**D**) and lung weight (**E**) in mice treated with NAM for four weeks, beginning either at 3 days or 4 weeks post infection. Symbols represent biological replicates (*n* ≥ 7/ group), and the center, upper and lower hinges, and upper and lower whiskers represent the 2nd quartile, 3rd and 1st quartile, and ±1.5 times the interquartile range (3rd quartile − 1st quartile), respectively. Data are presented as individual values for biological replicates with the group mean ± SEM for panels (**B**, **C**). Statistical significance was determined by two-sided, two-sample *t*-test not assuming equal variance (**B**, **C**, **E**) or two-sided, two-sample Wilcoxon rank-sum test (**D**). \**p* < 0.05, \*\**p* < 0.01, \*\*\**p* < 0.001, \*\*\*\**p* < 0.0001; exact *p*-values for each comparison are listed in Supplementary Data 1. Source data are provided as a Source Data file.

directed drug, isoniazid, was reversed by the respective inhibitor (Fig. 6B, C). This indicates that the effect of NAM depends on the maintenance of glycolytic flux through its conversion to NAD⁺. Additionally, FK866 and 2DG significantly increased the bacterial burden in the absence of any treatment, further indicating that BMDMs require NAD(H) and glycolytic flux for host protection at baseline.

Finally, we evaluated the suitability of NAM as a treatment for TB in the mouse model of TB. Mice were infected (~100 CFU) via the aerosol route and treated for four weeks starting at either 3 dpi (post-exposure prophylaxis) or 28 dpi (treatment). Both prophylactic and therapeutic approaches with NAM resulted in a statistically significant, ten-fold reduction in the bacterial burden in the lungs of mice (Fig. 6D), as well as a significant decrease in lung weight reflecting reduced inflammation (Fig. 6E).

In summary, our in vitro data show that NAM enhances macrophage control of *Mtb* infection specifically by its conversion to NAD⁺ through the NAD⁺ salvage pathway and its subsequent potentiation of glycolysis. Notably, our in vivo studies illustrate that orally administered NAM is effective as prophylaxis and treatment for *Mtb* infection. Overall, our findings point to NAM as a potential HDT for the treatment of TB.

## Discussion
The major conclusion of this study is that NAD(H) homeostasis in myeloid cells, maintained by LDHA activity and NAD⁺ salvage, supports glycolytic capacity and, subsequently, host protection in TB (Fig. 7).

Importantly, the clinical relevance of this work is rooted in the spatial and cellular distribution of LDHA in the spectrum of human TB lesions. Our in vivo experiments subsequently demonstrated that maximal glycolytic capacity in myeloid cells supports the induction of protective immunity, likely through the antimycobacterial effects of IFNγ. Our bioenergetic and pharmacological inhibition experiments show that *Mtb* disrupts glycolytic flux in infected macrophages by reducing the total abundance of the NAD(H) pool. We demonstrate that this virulence strategy can be countered through administration of the NAD⁺ precursor, NAM, which is a compound with known efficacy in TB. Our study represents a significant advance over previous work in the TB field that relies solely on 2DG or disruption of *Hif1A*, which not only suppress glycolysis, but also other essential pathways. Overall, our findings provide fundamental insights into how *Mtb* reprograms host glycolysis and highlight this pathway as a potential therapeutic target to control TB disease.

This study affirms our central hypothesis that glycolytic flux in myeloid cells is essential for host protection in TB. Importantly, the use of *Ldha*^LysM−/− mice extends previous studies that associate glycolytic myeloid cells and host protection by applying a more selective approach to the inhibition of glycolytic flux specifically in this subset of immune cells, rather than using an inhibitor of glucose uptake, 2DG[15,17,18], or HIF1α[15,18]. Since glycolysis requires NAD⁺ for the oxidation of glyceraldehyde-3-phosphate to 1,3-bisphosphoglycerate, and NAD⁺ can be regenerated through the reduction of the glycolytic end product, pyruvate, by LDHA, the target of our enzymatic manipulation is

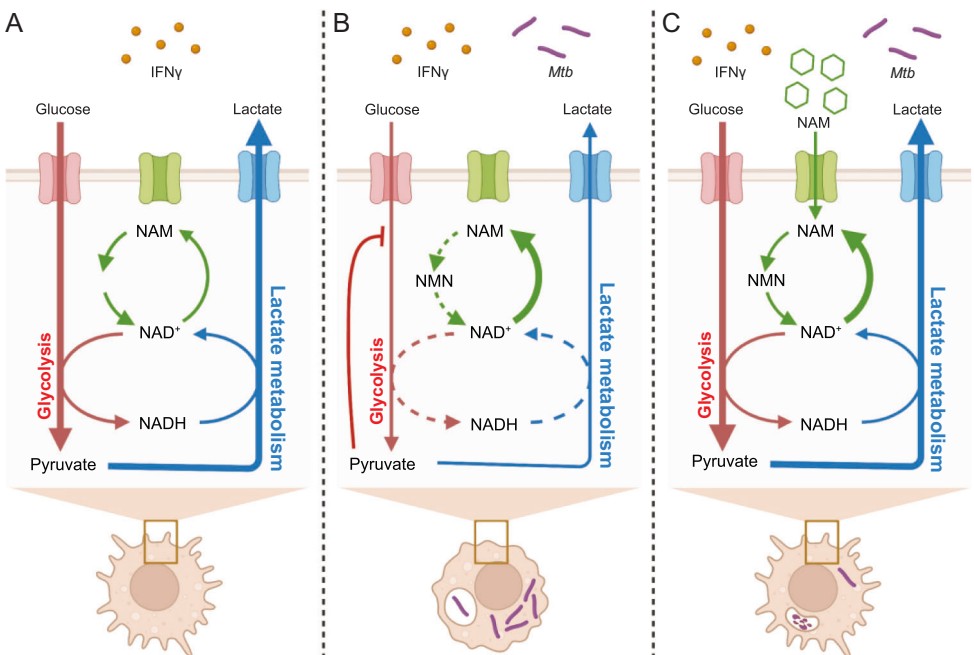

**Fig. 7 | NAD(H) homeostasis protects the host in TB.** Summary graphic depicting the proposed model for the metabolism of macrophages responding to stimulation with IFNγ, how *Mtb* disrupts this response, and how NAM restores it. **A** Following stimulation with IFNγ, macrophages exhibit high rates of glycolysis and lactate metabolism, necessitating the rapid cycling between NAD$^+$ and NADH. Under these conditions, the NAD$^+$ salvage pathway effectively maintains the pool of NAD(H). Together, this metabolic response enables the functional activation of the macrophage by IFNγ. **B** In *Mtb*-infected macrophages, the metabolic response to IFNγ is prevented by disrupting NAD(H) homeostasis. This may be due to decreased synthesis or increased NAD+ consumption. Regardless, reduced NAD(H) availability impairs the conversion of pyruvate to lactate, leading to an accumulation of pyruvate. Pyruvate is then converted to citrate which inhibits glycolytic flux. The functional consequence is an impairment in the host-protective effects of IFNγ. **C** Following treatment with NAM, *Mtb*-infected macrophages can respond effectively to IFNγ, as the NAM is metabolized to NAD$^+$. While the insult to NAD(H) homeostasis is likely still present, NAM supplementation addresses the immediate downstream impact of this insult by supporting NAD$^+$ synthesis. Consequently, the metabolic and functional responses to IFNγ and other environmental signals is restored, ultimately leading to host protection in TB.

external to process of glycolysis altogether. Hence, we effectively used an endogenous regulatory strategy, i.e., NAD$^+$ availability, as a metabolic valve to modulate glycolytic flux for the control of *Mtb* disease.

Conditional and whole-body knockouts of the transcription factor HIF1α have been used to examine the role of glycolysis during *Mtb* infection[15,18]; however, the resultant pleiotropic effects confound interpretation of these results. Based on the findings presented here, it is clear that inhibiting glycolysis, and more specifically inhibiting LDHA, is not a suitable approach to HDT for TB. Interest in this approach has been based largely on the association between glycolysis, inflammation, and TB disease[3,14,49]. However, our in vivo findings suggest that while reduced glycolytic capacity in myeloid cells may reduce inflammation in the short term, it ultimately leads to chronic inflammation, worse pathology, and worse survival. Our data may seem to conflict with a report that *Mtb*-infected mice treated with FX11, a NADH-analog inhibitor of LDH, showed modest improvements in bacterial burden and pathology[50]. However, these results could be explained by (i) a short-term reduction in inflammation that is ultimately detrimental to the host or (ii) by a mechanism independent of LDH inhibition, as the on-target effect of FX11 could not be confirmed in vivo.

Following our initial in vivo experiments, we focused our investigation in primary murine macrophages to the response to IFNγ, the transcriptional response most dysregulated as determined by our unsupervised analysis of 50 major cellular pathways and supported by the TB literature[15,18,19]. Similarly, we focused our studies on macrophages to best complement this literature, as well as studies of *Mtb*-induced glycolytic defects[16,19,20], all of which center on macrophages. There are, however, notable differences in the immunometabolism of murine and human cells. Specifically, classical activation of murine

macrophages involving stimulation with IFNγ is associated with NO-driven reprogramming of metabolism toward a glycolytic phenotype[51]—a pathway not fully evident in human cells. Our findings in hMDMs in the absence of IFNγ, however, demonstrate that the protective effects of maintaining NAD(H) homeostasis extend beyond the unique immunometabolism of murine macrophages.

Similarly, the reduction of glycolytic capacity due to NAD(H) depletion following *Mtb* infection is likely sufficiently generalizable to disrupt a wide array of glycolysis-dependent functions, including, most proximally, responses to immunomodulatory signals such as pathogen-associated molecular patterns, damage-associated molecular patterns and other cytokines[26]. Specifically, NAD$^+$ salvage is essential for polarization of murine macrophages to a glycolytic, inflammatory phenotype. Thus, our findings link this observation with the repeated observation that heat-killed or irradiated *Mtb* induces a robust glycolytic response while virulent *Mtb* does not. In other words, while the physical components of the bacillus are sufficient to induce a robust glycolytic response, viable virulent *Mtb* suppresses that very response[16,19,22]. Our data in *Ldha*$^{LysM-/-}$ mice further suggest that disruption of glycolysis in infected phagocytes may underlie the known ability of *Mtb* to delay the induction of adaptive immunity, as evidenced by the dramatically delayed immune response in these mice. Thus, while disruption of the central pathways of NAD(H) homeostasis and glycolysis likely affects various effector functions of macrophages, these varied effects are likely unified through the intermediate disruption of polarization to an inflammatory phenotype.

Our findings raise the following question: how does *Mtb* deplete NAD(H) levels? Although outside the scope of the study, this effect, and why it is limited to virulent *Mtb*, can be partly explained by the secretion of tuberculosis necrotizing toxin (TNT), a NAD$^+$

glycohydrolase[52]. WT *Mtb* was shown to significantly reduce NAD+ abundance in infected macrophages; however *Mtb* lacking functional TNT still reduced NAD+ to around 50% of WT *Mtb*[41,42]. This discrepancy appears to be explained by the observation that macrophages treated with lipopolysaccharide, a constituent of the outer membrane of gram-negative bacteria, exhibit a similar reduction in intracellular NAD(H) and increased reliance on the NAD+ salvage pathway[26]. In this context, glycolysis was not affected unless NAD+ salvage was also inhibited. Thus, NAD(H) depletion and inhibition of glycolysis may result from both an increased reliance on NAD+ salvage and increased strain on this pathway due to NAD+ degradation by TNT (Fig. 7B).

Given the downstream consequences of NAD(H) depletion and inhibition of glycolysis, we supplemented macrophages with NAM to restore NAD(H) homeostasis; however, interpretation of this effect on the restriction of bacterial replication is limited by the known effect of NAM as an antimycobacterial compound[43–45]. Specifically, NAM predominantly exerts a direct antimycobacterial effect dependent on bacterial pyrizinamidase (PncA), though recently it has also been shown to exert a PncA-independent, host-directed effect[47,48]. While we do not distinguish between these effects in our study, we show that the overall antimycobacterial effect of NAM was dependent on glucose uptake and NAD+ salvage in the host. These findings further detail the host-dependent mechanism of this long-studied compound. Moreover, despite this limitation, the increased bacterial burden observed in LDHA-deficient macrophages and macrophages treated with the inhibitor of NAD+ salvage, FK866, directly implicate NAD(H) homeostasis in bacterial control.

Despite its relatively early discovery, NAM appears to have been abandoned as a therapy during the golden age of antibiotic discovery for TB given its apparent redundancy with pyrazinamide as well as the publication of a brief report suggesting it antagonizes isoniazid when used in combination[53,54]. While this may be true, the landscape of TB has shifted dramatically in the last 60 years, with an increase in the incidence of TB to over 10 million new cases annually and the development of resistance to the frontline drugs that displaced NAM. Here, we provide further evidence of a host-dependent effect of NAM, metabolic requirements for its activity, and a modern-day demonstration of its efficacy as a treatment for TB using two treatment regimens in vivo. Logistically, it satisfies many of the criteria for an optimal novel TB treatment regimen set forth by the WHO, given that it is inexpensive, orally bioavailable, shelf stable, and remarkably safe and tolerable. Finally, it is well studied and routinely used in humans for various indications[54–61]. Ultimately, these characteristics make NAM appealing as an old tool in a modern setting.

Further limitations of our study include that we primarily considered *Ldha* deletion as a specific approach to the NAD(H)-mediated manipulation of glycolysis. However, when pyruvate is a substrate, lactate is the product, and lactate has a variety of immunomodulatory effects in and beyond TB[31,62,63]. Given that we used a myeloid-specific knockout mouse, and lymphocytes make up the vast majority of cells within the lungs of both WT and *Ldha^LysM−/−* mice at all time points surveyed, we consider it unlikely that lactate played a greater role than the glycolytic defect we directly observed in myeloid cells.

In summary, we demonstrate the importance of two complimentary pathways responsible for the maintenance of NAD(H) homeostasis, lactate metabolism and NAD+ salvage, for host protection mediated by myeloid cells in TB. We observe that *Mtb*-infected macrophages exhibit an increased reliance on NAD+ salvage, which we expand to connect independent observations that *Mtb* infection diminishes NAD+ levels in macrophages[41,42], *Mtb* decreases glycolytic capacity in infected macrophages[16,19,20], and the NAD+ precursor NAM exerts a host-directed antimycobacterial effect[43–45,48]. Finally, we validate the antimycobacterial effect of NAM in an animal model of TB, providing rationale for its use as a new approach to TB prophylaxis or treatment.

## Methods

### Ethics statement for human research

The study of human lung pathology was approved by the University of KwaZulu-Natal Biomedical Research Ethics Committee (BREC), Class approval study number BCA 535/16. Patients undergoing lung resection for TB, their study protocol, associated informed consent documents, and data collection tools were approved by the UKZN BREC (Study ID: BE 019/13). Written informed consent was obtained from patients recruited from King DinuZulu Hospital Complex, a tertiary center for TB patients in Durban, South Africa.

Collection of human blood was approved by the Institutional Review Board of the University of Alabama at Birmingham. Written informed consent was received from all participants prior to inclusion in the study.

### Ethics statement and husbandry for animal research

*LysM^+/+Ldha^fl/fl* (*Ldha^fl/fl*) and *LysM^+/creLdha^fl/fl* (*Ldha^LysM−/−*) mice on a C57BL/6 background were originally sourced from an independent investigator[31] and maintained in pathogen-free facilities. C57BL/6 J mice were obtained from the Jackson Laboratory. All studies used sex-matched, age-matched, littermates of both sexes. Littermates within each sex were cohoused until the start of experiments. For in vivo experiments, mice were aged 8–12 weeks old at the start, while, for the ex vivo generation of BMDMs, mice aged 8–16 weeks old were used. At the start of in vivo experiments, mice were housed under ABSL-3 conditions and monitored daily. Housing conditions: light/dark cycle: 12 h/12 h with lights on from 7AM–7PM. Ambient temperature approximately 72 F with relative humidity of approximately 50%. All procedures and protocols were approved by the Institutional Animal Care and Use Committee of the University of Alabama at Birmingham.

### Bacteria

*Mtb* H37Rv or luciferase-expressing *Mtb* H37Rv (*Mtb*-lux) was used in all experiments. The *Mtb*-lux strain used in this study was generated in our laboratory by electroporating laboratory strain *Mtb* H37Rv with the plasmid construct pMV306hsp+Lux created in the labs of Brian Robertson and Siouxsie Wiles and procured from Addgene (cat # 26159)[64]. Liquid cultures of *Mtb* were grown at 37 °C with shaking in Middlebrook 7H9 (Thermo Fisher Scientific, cat # DF0713-17-9) broth supplemented with 0.02% tyloxapol and albumin (Millipore Sigma, cat # 3116956001), Dextrose (Thermo Fisher Scientific, cat # DF0155-17-4), and saline (ADS) while solid cultures were grown at 37 °C standing on Middlebrook 7H11 agar (Thermo Fisher Scientific, cat # L12203) supplemented with ADS and 0.2% glycerol. Stocks of *Mtb* were generated from mid-log phase liquid cultures by 1:1 dilution with 50% glycerol in water and stored at −80 °C for up to 6 months.

### Immunohistochemistry

Human lung samples (Table S1) were aseptically removed and fixed in 10% neutral buffered formalin (10% NBF). These samples were processed in a vacuum filtration tissue processor using a xylene-free protocol. Tissue sections were embedded in paraffin wax. Human lung tissue was cut into 2 μm-thick sections, mounted on charged slides, and heated at 56 °C for 15 min on a hotplate. Mounted sections were dewaxed in 2 changes of xylene followed by rinsing in 2 changes of 100% ethanol and 1 change of SVR (95%). Slides were then rinsed in tap water for 2 min followed by antigen retrieval via Heat Induced Epitope Retrieval (HIER) in trisodium citrate (pH = 6.0) for 30 min. Slides were cooled for 15 min and rinsed in tap water for 2 min. Endogenous peroxide activity was blocked by exposing tissue sections to 3% hydrogen peroxide (Leica Novolink) for 10 min at room temperature (RT). Slides were then rinsed in PBST and blocked with protein block (Leica Novocastra) for 5 min at RT. Sections were incubated with primary antibody for lactate dehydrogenase (LDHA ab125683, abcam,1:500) followed by rinsing in PBST and incubated with the appropriate

polymer (Leica Novolink) for 30 min at RT. Slides were then rinsed and stained with Diaminobenzidine (DAB) for 5 min, rinsed under running water, and counterstained with hematoxylin for 2 min. Slides were rinsed in tap water, blued in 3% ammoniated water for 30 s, rinsed in tap water, dehydrated in ascending grades of alcohol, cleared in xylene, and mounted with DPX (Distyrene, Plasticizer, and Xylene). For isotype control sections, IgG4 (LSBio cat # LS-C70325) was used in place of the same concentration/dilution as the primary antibody. Tissue staining and staining controls were performed at least three times independently post optimization. Slides were scanned using a Hamamatsu NDP slide scanner (Hamamatsu NanoZoomer RS2, Model C10730-12) and its viewing platform software (NDP.View2).

### Ex vivo experiments

For the generation of BMDMs, mice were euthanized according to institutional protocols. Femurs and tibias were isolated, and the bone marrow was flushed out through a cell strainer (70 μM pore size) with PBS supplemented with 5% FBS. Red blood cells were lysed with ACK lysis buffer, and the residual bone marrow cells were plated in tissue-culture-treated microplates at a density of $2 \times 10^5$ cells per well in RPMI 1640 containing 2 mM glutamine (Thermo Fisher Scientific, cat # 21875034), 10% FBS (Thermo Fisher Scientific, cat # 10082147), 10 mM HEPES, 20 ng/mL murine M-CSF (BioLegend, cat # 576404), and penicillin (100 units/mL) and streptomycin (100 units/mL; AB/AM). Cells were differentiated in this media for 6 days. Media was changed once during this time, on day 3.

For the generation of hMDMs, blood was collected from healthy volunteers in EDTA-containing vacutainers (BD, cat # 02-683-99 C). Blood was mixed with 1 volume of PBS, and the mixture was gently layered over one volume of Histopaque 1077 (Millipore Sigma, 10771) and spun at 400 g for 30 min. Peripheral blood mononuclear cells were isolated from the plasma-histopaque interface. Monocytes were isolated from PBMCs by negative selection as per the manufacturer's instructions and plated in 96-well microplates at a density of $9 \times 10^4$ cells per well in RPMI 1640 containing 2 mM glutamine, 10% FBS, 10 mM HEPES, 20 ng/mL GM-CSF (BioLegend, cat # 572903), and AB/AM. Monocytes were differentiated in this media for 6 days with one media change on day 3.

After 6 days, BMDMs and hMDMs were infected as described below and exposed overnight (-18 h) in RPMI 1640 containing 2 mM glutamine, 10% FBS, 10 mM HEPES and either 5 ng/mL M-CSF for BMDMs or no cytokine for hMDMs (complete media) and the indicated treatment condition(s): IFNγ stimulation (10 ng/mL; BioLegend, cat # 575304); supplementation with pyruvate or nicotinamide (NAM) (1 mM; Millipore Sigma); treatment with the LDHA/LDHB inhibitor, GSK 2837808A (5 μM; Tocris, 5189) or the NAMPT inhibitor FK866 (200 μM; Millipore Sigma, cat # F8557); and/or *Mtb* infection at a multiplicity of infection (MOI) of 5:1 (unless otherwise indicated). For *Mtb* infections, *Mtb* was quantified by optical density at a wavelength of 600 nm ($OD_{600}$), assuming an $OD_{600}$ of 1 corresponds to -$10^8$ CFU. The required amount of liquid culture in the mid-log phase of growth or glycerol stock was passaged 5 times through a 27-gauge needle and added to complete RPMI (infection media). BMDMs were incubated with infection media at 37 °C and 5% $CO_2$ for 4 h, after which BMDMs were washed 3 times in PBS and returned to complete RPMI containing any other treatment conditions indicated within the experiment.

### Quantification of LDHA protein expression

BMDM protein lysates were quantified by BCA analysis (Thermo Fisher Scientific, cat # 23225), diluted to a concentration of 1 mg/mL, and mixed 1:1 with 2x Laemmli Buffer (Bio-Rad Laboratories, cat # 161-0737). 10 μg of protein was loaded from each sample into separate wells of a 4–15% polyacrylamide gel (Bio-Rad Laboratories, cat # 456-8084). Electrophoresis was performed in 1x Tris/Glycine Buffer (Bio-Rad Laboratories, cat # 1610734) at 100 V, and protein was transferred

to a PVDF membrane (Bio-Rad Laboratories, cat # 1620177) overnight at 35 V and 4 °C. Following the transfer, non-specific protein binding was blocked by incubation of the membrane in 5% non-fat milk for 30 min. The membrane was then incubated with the primary anti-LDHA antibody (ProteinTech, cat # 19987-1-AP) diluted 1:1000 in 5% non-fat milk at RT for one hour. The membrane was washed three times in TBST and incubated with the HRP-conjugated goat-anti-rabbit secondary antibody (Kindle Biosciences, cat # R1004) diluted 1:1000 in 5% non-fat milk at RT for one hour. The membrane was then covered in KwikKwant HRP Substrate Solutions A + B (Kindle Biosciences, R1004) and images on a KwikKwant Imager (Kindle Biosciences). Images were quantified with the Fiji/ImageJ[65] gel analysis tool, version 1.52p.

### LDH functional assay

LDH enzyme activity was determined by measuring the conversion of NADH to NAD$^+$ in the presence of saturating concentrations of NADH and pyruvate. 10 μg of protein from separate BMDM cell lysates was loaded into separate wells and incubated in a final volume of 200 μL of PBS containing 10 mM pyruvate and 300 μM NADH (Acros Organics, cat # 271102500) in an ultraviolet-transparent 96 well microplate (Thermo Fisher Scientific, cat # 8404). NADH depletion was measured by the absorbance at 340 nm. To determine the rate of reaction, we performed a series of kinetic measurements with the minimal read time for the plate in the Synergy H1 Hybrid Multi-Mode Reader and Gen5 software (BioTek).

### Morphometric analysis

The open-source software QuPath version 0.4.3 (https://qupath.github.io)[34] was used for morphometric analysis of whole-slide images. Stain vectors were adjusted for each image for optimal segmentation. Automated cell segmentation was performed for all tissue sections within using the "cell detection" tool following optimization of the segmentation parameters across representative sections from each experimental group. To determine the local cellular density measurement for each cell, we employed a modified version of the "create counts map.groovy" script published by the QuPath developer (https://gist.github.com/petebankhead). Briefly, the images were arbitrarily divided into many smaller regions (pixels) of equal size, and the number of nuclei within each region was determined. Pixels containing the edge of the tissue section were excluded from analysis. To improve on the discrete nature of this approach, the count of each pixel was adjusted to the mean values of its neighbors by applying a gaussian blur filter. The resulting value for each pixel was then assigned to the nuclei within that region. Data were exported and analyzed in R statistical software (see "Statistical analysis" section).

### Aerosol infection with Mtb

Mice within each experiment were infected as a cohort with *Mtb* H37Rv using an aerosol inhalation exposure system (Glas-Col) within ABSL-3 containment. *Mtb* was obtained from mid-log phase liquid cultures, washed once in PBS via centrifugation, and resuspended in 6 mL of PBS at an $OD_{600}$ of 0.07. This suspension was then transferred to the nebulizer of the exposure system, allowing for exposure of the mice to aerosolized *Mtb*. 24 hpi, 3–4 mice were euthanized and the initial bacillary burden was determined as described in the section "Quantitation of bacterial burden."

### Quantitation of bacterial burden

At indicated timepoints, mice were euthanized according to institutional protocols. Lungs and spleens were dissected and homogenized in 2 mL of PBS and serially diluted in PBS. 100 μL of appropriate dilutions was plated on 7H11 agar as described in the section "Bacteria." Colonies were counted after 21 days, and the total CFU per organ was determined by normalizing counts to dilution factor, volume plated, volume of the homogenate, and the proportion of the total organ plated.

## Flow cytometry

For flow cytometric analysis, the following anti-mouse antibodies and stain were acquired from BioLegend: PerCP-Cy5.5 anti-Ly6C (cat # 128012; 1:200), APC-Cy7 anti-CD11b (cat # 101226; 1:200), AF700 anti-Ly6G (cat # 127622; 1:500), PE-Cy7 anti-F4/80 (cat # 123114; 1:200), BV785 anti-CD11c (cat # 117336; 1:500), BV605 anti-CD45 (cat # 103155; 1:100), BV421 anti-CD64 (cat # 139309; 1:67), BV650 anti-I-A/I-E (cat # 107641; 1:200), PE anti-CD24 (cat # 138503; 1:200), and Zombie Aqua fixable viability dye (cat # 423101).

Mice were euthanized at indicated timepoints according to institutional protocols. Immediately following sacrifice and thoracotomy, the pulmonary vasculature was perfused with PBS via the right ventricle. Lobes taken for cytometric analysis were minced in Dulbecco's Modified Eagle Medium and incubated with Liberase (Roche) 2 mg/mL at 37 °C for 30 min. The resulting cell suspension was passed 5 times through a 20-gauge needle, 5 times through a 23-gauge needle and filtered through a cell-strainer with a 40 μm pore size. Cells were washed once in PEB buffer and stained with fixable live/dead stain for 30 min on ice. Cells were washed in PEB and stained for the indicated surface markers by staining cells with fluorophore-conjugated antibodies for 30 min at 4 °C. Cells were then fixed in 4% paraformaldehyde for 30 min, resuspended in PBS, and stored at 4 °C in the dark until they could be analyzed the following day. Flow cytometry acquisition was performed using an Attune NxT cytometer (Attune cytometric software version 5.2; Thermo Fisher Scientific), and analysis was performed with FlowJo software version 10 (Tree Star).

## RNA sequencing and analysis

RNA was isolated from the lungs of *Mtb*-infected mice stored in RNA-later (Thermo Fisher Scientific, cat # AM7024) at −80 °C using the RNeasy plus Mini Kit (Qiagen, cat # 74134) according to the manufacturers protocol. Briefly, lung tissue was homogenized in RLT lysis buffer and passed through a genomic DNA eliminator column. The eluate was mixed with an equal volume of 70% ethanol and passed through an RNA-binding column. The column was washed once with RW1 buffer and twice with RPE, following which the membrane was thoroughly dried and the RNA was eluted in nuclease-free water.

mRNA sequencing was then performed on the Illumina Next-Seq500 as described by the manufacturer using NextSeq software version 4.2.0 (Illumina, Inc). RNA quality was assessed using the Agilent 2100 Bioanalyzer. RNA with a RNA Integrity Number (RIN) of ≥7.0 was used for sequencing library preparation. RNA passing quality control was converted to a sequencing ready library using the NEBNext Ultra II Directional RNA library kit with poly-A selection per the manufacturer's instructions (New England Biolabs). The cDNA libraries were quantitated using qPCR in a Roche LightCycler 480 with the Kapa Biosystems kit for Illumina library quantitation (Kapa Biosystems) prior to cluster generation.

STAR (version 2.7.3a) was used to align the raw RNA-Seq fastq reads to the GRCm38 p6 Release M24 reference genome from Gencode[66]. Following alignment, HTSeq-count (version 0.13.5) was used to count the number of reads mapping to each gene[67]. Normalization and differential expression was then applied to the count files using DESeq2[68]. Further analysis of differentially expressed genes was performed as described in the "Statistical analysis" section.

## Multiplexed cytokine analysis

Protein was isolated from the lungs of *Mtb*-infected mice stored in RNAlater at −80 °C by homogenization in RIPA buffer followed by 3 freeze-thaw cycles. Lysates were then centrifuged to remove debris, and the resulting supernatant was passed through a 0.22 μm spin column to removed debris and *Mtb*. The quantity of total protein in each sample was determined with the Pierce BCA Protein Assay Kit and each sample was diluted to 500 μg/mL.

Cytokine quantitation was performed with the Bio-Plex Pro Mouse Cytokine 23-plex Assay (Bio-Rad, cat # M60009RDPD) according to the manufacturer's instructions. In brief, samples were incubated with magnetic-bead-bound capture antibodies on a shaking platform at 850 RPM for 30 min followed by two washes in Bio-Plex wash buffer with a hand-held wash station. The process was repeated with biotinylated detection antibodies. Finally, PE-conjugated streptavidin was added to samples for a 10-minute shaking incubation. Samples were washed three times in Bio-Plex wash buffer, resuspended in assay buffer, and processed on the MAGPIX instrument platform with xPONENT software version 5.1.0 (Luminex).

## Extracellular flux analysis

Extracellular flux analysis was performed using a XFe96 Extracellular Flux Analyzer (Agilent; Wave software version 2.6.1.53), XFe96 Flux Packs (Agilent, cat # 102416-100), and effector reagents purchase from Millipore Sigma. BMDMs were generated and treated in XFe96 cell culture plates as described in the section "ex vivo experiments." A combination of the Glycolysis Stress Test and Mito Stress Test was performed as previously described[39]. Cells are incubated in glucose-free RPMI in a non-$CO_2$ incubator the hour preceding the run. Initial ECAR measurements are used to determine non-glycolytic acidification. The first injection of glucose (final concentration: 10 mM) is followed by measurements to determine basal respiration (OCR) and glycolytic acidification (ECAR). The second injection of oligomycin (final concentration: 1.5 μM) is followed by measurements to determine ATP production by oxidative phosphorylation (OCR) and glycolytic capacity (ECAR). The third injection of FCCP and pyruvate (final concentrations: 1.5 μM and 1 mM, respectively) is followed by measurements to determine maximal respiration (OCR) and the spare respiratory capacity (OCR). Finally, the fourth injection of rotenone and antimycin A (final concentrations: 2.5 μM and 1.25 μM, respectively) is used to determine the non-mitochondrial oxygen consumption (OCR).

Additionally, the final injection also contained Hoescht 33342 (final concentration: 2 μg/mL; Thermo Fisher Scientific, cat # 62249) to stain cell nuclei. Cell counts in each well were then determined with whole-well fluorescent imaging using a Cytation 5 Cell Imaging Multi-Mode Reader and Gen5 software, version 3.12.08 (BioTek). In brief, images were acquired with the DAPI filter cube, 4x objective, and laser autofocus; stitched using the linear blend method; and processed using a combination of background flattening, blurring, and optimization of cellular analysis parameters. In instances of significant cell clumping (i.e., bacterial infection), cell count was determined by comparing mean fluorescent intensity to wells with an even distribution of cells (i.e. uninfected). ECAR and OCR values within each well were then normalized per 10,000 cells.

Bioenergetic parameters were determined from the ECAR and OCR values obtained from the glycolysis and mitochondrial stress tests as described elsewhere[11,12,39]. Beginning with parameters from the glycolysis stress test: Non-glycolytic acidification was determined by the third ECAR value following the initiation of the experiment (the ECAR value immediately preceding the injection of glucose). Glycolytic acidification was determined by subtracting non-glycolytic acidification from the third ECAR value following the injection of glucose (the ECAR value immediately preceding the injection of oligomycin). Glycolytic capacity was determined by subtracting non-glycolytic acidification from the third ECAR value following the injection of oligomycin (the ECAR value immediately preceding the injection of FCCP/pyruvate). Glycolytic reserve was determined by subtracting the value determined for glycolytic acidification from the value determined for glycolytic capacity.

Non-mitochondrial respiration was determined by the third OCR value following the injection of Rotenone/Antimycin A (the final OCR value in the experiment). Basal respiration was determined by

subtracting non-mitochondrial respiration from the third OCR value following the injection of glucose (the OCR value immediately preceding the injection of oligomycin). Proton leak was determined by subtracting non-mitochondrial respiration from the third OCR value following the injection of oligomycin (the OCR value immediately preceding the injection of FCCP/pyruvate). ATP production was determined by subtracting proton leak from basal respiration. Maximal respiration was determined by subtracting non-mitochondrial acidification from the third OCR value following the injection of FCCP/pyruvate (the OCR value immediately preceding the injection of rotenone/antimycin A). Spare respiratory capacity (SRC) was determined by subtracting basal respiration from maximal respiration.

### Targeted metabolomics
Bone marrow cells were seeded in a 6-well microplate at a density of $3 \times 10^6$ cells per well and differentiated as described in the section "ex vivo *experiments.*" Following the indicated treatment condition, cells were washed once in warm, glucose-free RPMI 1640 and incubated at 37 °C and 5% $CO_2$ for 2 h in glucose-free RPMI (Thermo Fisher Scientific, cat # 11879020) supplemented with 10 mM $^{13}C_6$-glucose (Cambridge Isotope Laboratories, cat # CLM-1396-2) and dialyzed FBS (Thermo Fisher Scientific, cat # 26400044). Following this incubation, cells were quickly washed three times with warm PBS and lysed by the addition of ice-cold extraction buffer (50% methanol in water) spiked with 20 ng/mL deuterated succinate (Millipore Sigma, cat # 293075-1 G) followed by scraping. Three technical replicate wells were pooled for each sample. Samples were subjected to three freeze/thaw cycles and then filtered through a 0.22 μm spin column. Protein content was estimated by the BCA assay, and samples were dried under vacuum for subsequent transport on dry ice. For liquid chromatography tandem mass spectrometry (LC-MS/MS), dried samples were reconstituted in 150 μL of water and filtered through a 0.22 μm spin column. 50 μL from each sample was mixed 1:1 with acetonitrile spiked with deuterated alanine for quantitation of amino acids. Standards for all metabolites analyzed were purchased (Millipore Sigma) and used as positive controls. Relative quantities of each metabolite were normalized to the internal standard, which is deuterated succinate for central carbon metabolites and deuterated alanine for amino acids, and to the protein concentration for that sample. Data were processed using Skyline version 4.1.0 software (MacCoss Lab Software).

### Glucose uptake assay
Glucose uptake rates were determined with the Glucose Uptake-Glo Assay (Promega, J1341) according to the manufacturer's instructions. Following the indicated treatment, BMDMs were washed in PBS and incubated with 1 mM 2DG for 10 min, allowing for 2DG uptake and phosphorylation to 2DG-6-phosphate. After this incubation, stop buffer was added to each well followed by neutralization buffer. Finally, the 2DG-6-phosphate detection reagent was added, and samples incubated for 30 min at RT. Luminescence was quantified with an integration time of 1 s using a Cytation 5 Cell Imaging Multi-Mode Reader (BioTek).

### NAD$^+$/NADH quantification
NAD$^+$ and NADH were quantified using the NAD/NADH-Glo Assay (Promega, G9071) according to the manufacturer's instructions. Cells were generated and treated as described in "Ex vivo experiments". They were then washed once with PBS, and media was replaced with 50 μL of PBS. 50 μL of 0.2 N NaOH with 1% CTAB was added to each well, and each sample was split into two 50 μL aliquots. To one group of aliquots, 25 μL of 0.4 N HCl was added, and both sets of samples were incubated for 15 min at 60 °C and 10 min at RT. 25 μL of 0.5 M Tris base was added to the HCL treated samples, while 50 μL of a 1:1 mixture of the HCl and tris solutions was added to untreated samples. 100 μL of NAD + /NADH-Glo detection reagent was then added to each well, and

samples were incubated for 30 min at RT. Following incubation, luminescence was quantified with an integration time of 1 s using a Cytation 5 Cell Imaging Multi-Mode Reader (BioTek).

### Administration of compounds to mice
Nicotinamide (NAM) was administered to mice seven days per week via the drinking water at a concentration of 0.8% w/v, approximating a dose of 1 g/kg body weight/day. Therapeutic approaches consisted of a post-exposure prophylaxis model in which treatment was started at 3 days post infection and continued until sacrifice at 4 wpi, and a model for treatment of established infection in which treatment was started at 4 wpi and continued until 8 wpi.

### Statistical analysis
All statistical analyses were performed with R statistical software (version 4.0.2)[69] in the RStudio integrated development environment (version 1.3)[70]. Unless otherwise specified, data were organized and graphed with the "tidyverse" collection of R data science packages[71]. Heatmaps and hierarchical clustering was performed using the "pheatmap" package version 1.0.12[72]. Gene set enrichment analysis was performed with the "fgsea" package to test for the enrichment of each of the 50 hallmark gene sets from the MSigDB, which collectively identify many of the most well-defined biological processes, among our differentially expressed genes ranked by fold-change[73]. Dose-response curves and linear regressions were calculated using the "stats" package.

### Authorship inclusion and ethics
This study was conducted with the support of an international team of investigators. Specifically, the intellectual and practical contribution of the investigators located at institutions in Durban, South Africa have ensured the relevance of this research to both locally and more broadly to regions with a high burden of tuberculosis. Appropriate parts of the study were approved by local ethics review committees as described in the Methods section. Biohazardous experiments were conducted at the highest standard considered across participating sites. Throughout this study, efforts were made to avoid stigmatization of participants and, more broadly, people with tuberculosis. Finally, the interests of many groups−patients, clinicians, community members, funders, and investigators−were considered in guiding the direction of this study to promote the equitable distribution of its potential benefits.

### Reporting summary
Further information on research design is available in the Nature Portfolio Reporting Summary linked to this article.

## Data availability
The RNA sequencing data generated in this study have been deposited in the NCBI GEO Repository under accession code GSE234115 [ncbi.nlm.nih.gov/geo/]. Other data generated in this study are provided in the Supplementary Information. Source data are provided with this paper.

## Code availability
Scripts for data analysis were written in the R statistical programming language using freely available code as described in the "Materials and Methods" section. No original code or algorithms were generated for this study.

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

## Acknowledgements

This work was supported by NIH grants R01Al111940, R01AI134810, R01AI137043, R33AI138280, R21A127182, P30 DK079337, and pilot funds from the UAB CFAR and UAB Heersink School of Medicine to AJCS. This study was funded, in part, through Wellcome Strategic Core award 201433/Z/16/A with a CC BY public copyright license. Support was also received from the South African (SA) Medical Research Council, and a SA NRF BRICS Multilateral grant to AJCS. The authors wish to thank Drs. Pankaj Seth and Barbara Wegiel (Harvard Medical School) for providing $Ldha^{LysM-/-}$ mice and Dr. Sixto Leal (UAB) for assistance in establishing protocols for human blood collection. This work was supported by the staff, management, and resources of the UAB Bioanalytical and Redox Biology Core, Heflin Genomics Core, Comprehensive Flow Cytometry Core, and the Southeastern Biosafety Laboratory Alabama Birmingham (SEBLAB), a NIAID-supported (UC6 AI058599) Regional Biocontainment Laboratory.

## Author contributions

Conceptualization: H.T.P., A.A., A.J.C.S. Methodology, H.T.P., A.J.C.S. Formal analysis: H.T.P. Investigation: H.T.P., K.C.C., V.P.R., S.N., R.R.S., K.N., K.L., T.N., J.N.G. Visualization: H.T.P., A.J.C.S. Writing—Original Draft: H.T.P., A.J.C.S. Writing—Review & Editing: H.T.P., J.N.G., A.J.C.S. Supervision: A.A., A.J.C.S. Funding Acquisition: A.J.C.S. All authors discussed the results and commented on the manuscript.

## Competing interests

The authors declare no competing interests.
