## [Peer Review File · Nature Communications]

NAD(H) homeostasis underlies host protection mediated by glycolytic myeloid cells in tuberculosisREVIEWER COMMENTS

Reviewer #1 (Remarks to the Author):

This is an interesting paper, which improves our understanding of early events in TB-host biology. In carefully designed murine experiments, the authors do a detailed study of glycolysis disruption in mouse macrophages by *Mycobacterium tuberculosis*. They identify a depletion of NAD⁺ by Mtb as a central player in this process. In cellular and also in in vivo experiments, they go on to demonstrate the host benefit of rescuing this NAD⁺ depletion as a potential host directed therapy.

The following points are made which the authors may seek to address.

Although Figure 1, describes LDHA immunostaining in human tissue (was there an isotype control?); no further mechanistic experiments are performed using human cells. This may be an issue because there are some notable differences in immunometabolic pathways seen in human versus murine material. Specifically, after stimulation, murine cells undergo Warburg metabolism, whereas human myeloid cells are more likely to increase both glycolysis and oxidative phosphorylation after a while.

As the authors know, pathways of immunity to TB are different in human and mouse cells also. Mouse cells efficiently kill *Mycobacterium tuberculosis* after stimulation with interferon gamma and the production of nitric oxide. Such pathways are not evident in human cells. Palmiere et al (PMID 32019928) suggested that NO-driven reprogramming of metabolism should be considered a result rather than a mediator of inflammatory polarization – in murine cells. So, have they got any macrophage data using human cells?

The figures are complex but clear. Figures three in particular emphasizes the importance of early events in the mouse infection which dictate the final trajectory of the animal's health. In Fig 4, they use a LDH inhibitor which should be defined. In figure 5 they use a NADPH inhibitor which should be defined. In figure seven they demonstrate how NAM protects the host.

Maybe .. they can advise us on mechanism? in other words, how is NAD⁺ homeostasis linked to the ability of the macrophage to kill Mtb? Is it related to NO in mice? Autophagy or phagolysosome maturation events?

Reviewer #2 (Remarks to the Author):

The manuscript " NAD(H) homeostasis is essential for host protection mediated by glycolytic myeloid cells in tuberculosis"

This manuscript provides important insight into immunometabolic response of the host to *Mycobacterium tuberculosis* and specifically the role of glycolysis and the Warburg effect. Whether Mtb infection induces glycolysis has been confounded in the literature with as studies have found that it both enhances or represses glycolysis due to the factors discussed in the manuscript and differences in the model systems used. Here the authors data indicates that whilst Mtb does induce glycolysis the additional IFN γ stimulation of glycolysis is dampened by *M. tuberculosis* infection. This is shown with extensive experimental data from lungs resected from human TB patients, murine in vivo and ex vivo models of TB. The authors then explore the mechanisms behind this demonstrating that this is associated with NAD(H) homeostasis and go on to show that nicotinamide can be used as a host directed therapy. Whilst I have some minor issues this is an excellent study which is an important contribution to the field.

Specific comments

¹³C isotopologue profiling is performed within this study but the isotopologue profiles appear to be almost completely ignored, not described or discussed. This is important data and would be interesting to see also what happens to the rest of metabolism as these pathways don't act in silo.

The authors discuss differences in pool sizes to infer metabolic flux changes. Pool sizes cannot be used to do this, and this must be mentioned. The isotopologue data shows something slightly more complicated and interesting is going on with metabolism. In some of the labelling patterns whilst there is changes in pool size the isotopologue profile is unchanged indicating there is no change in the flux through the pathway. Indeed for some metabolites there is an increase in labelling suggesting increased flux at the top end of glycolysis with the lower end of glycolysis remaining unchanged presumably to conserve NAD. This looks like it is then driving flux through the PPP which would generate NADPH. This doesn't detract from the central message but if this data is presented it needs to be discussed.

The authors postulate that NADH availability is a potential mechanism of the observed phenotypes. As a minor point the addition of pyruvate did not change the NAD/NADH balance in the LDH deficient cells which seems to be a slight anomaly that needs explaining. The authors say its expected but that is slightly puzzling to me. Figure 5C authors also measure pool sizes to show that there is a build up of metabolites which could allosterically inhibit enzymes which is fine as long as there is no inference about metabolic flux here. There is lots of metabolites presented but only two discussed. Is there isotopologue labelling for this experiment to show how the flux changes in these conditions?

Minor points

This is an extremely well-written paper but I have a couple of minor suggestions below L93-94 "suggest the consequences of impaired glycolytic capacity in these cells" doesn't quite make sense. What are these consequences?

L422-425 "However, these results might be explained by a short-term reduction in inflammation, since the authors state that the effect of FX11 in vivo could not be confirmed as the on-target inhibition of LDH, leaving open the potential for other mechanisms for the protective effect." Perhaps reword as I don't quite understand this.

Reviewer #3 (Remarks to the Author):

This manuscript describes a substantial number of experiments that investigated the importance of NAD(H) homeostasis, glycolytic myeloid cells and, specifically, lactate dehydrogenase (LDH) for control of infections with Mycobacterium tuberculosis. The manuscript is well organized and well written.

The focus on LDH was nicely motivated by expression analyses of LDHA (representing one of two LDH subunits) in lung tissues resected from TB patients. Analyses of mice lacking LDHA confirmed that inactivation of LDHA indeed causes a (somewhat modest) increase in susceptibility to M. tuberculosis. These findings are supported by experiments with bone marrow derived macrophages (BMDMs), which demonstrate that LDHA-deficient macrophages are unable to respond metabolically to stimulation with IFN-gamma. Unfortunately, it remains to be demonstrate how LDHA-deficiency impacts replication of Mtb in BMDMs.

Several experiments were performed to demonstrate a link between NAD(H) homeostasis and glycolysis. That such a link exists is not surprising given the broad spectrum of enzymes that required NAD(H). What is surprising is the claim that NAD(H) homeostasis provides the means for a "highly selective manipulation of host-cell glycolytic flux". It's not clear to me what data substantiate this claim.

Finally, the authors demonstrate that nicotinamide (NAM) more potently inhibits growth of Mtb in BMDMs compared to standard liquid media and that oral NAM supplementation reduces the burden of Mtb in mice. These data confirm previous experiments and are not particularly surprising given the clinical data that were generated decades ago (see PMID 12567303 for a summary). However, I agree with the authors that NAM supplementation might be of clinical value in specific circumstances.

A central claim of the manuscript is that the impact of NAM supplementation is mediated by a "host-directed antimicrobial effect". Unfortunately, this claim is only weakly supported by

experimental data. This claim is also at odds with a previously report that failed to detect an impact of NAM on macrophages infected with Mtb containing a deletion in *pncA*, which renders Mtb resistant to the direct antibacterial effects of NAM. This report (PMID 31665359) is cited but the discrepancy in conclusions is not discussed and it's surprising that the authors did not utilize NAM resistant MTB to support one of their main conclusions.

Reviewer #1 (Remarks to the Author):

This is an interesting paper, which improves our understanding of early events in TB-host biology. In carefully designed murine experiments, the authors do a detailed study of glycolysis disruption in mouse macrophages by Mycobacterium tuberculosis. They identify a depletion of NAD⁺ by Mtb as a central player in this process. In cellular and also in in vivo experiments, they go on to demonstrate the host benefit of rescuing this NAD⁺ depletion as a potential host directed therapy.

Response: We thank the Reviewer for highlighting the strengths of the manuscript, including the depth of the investigation into host metabolism and *in vivo* experimentation.

The following points are made which the authors may seek to address.

Although Figure 1, describes LDHA immunostaining in human tissue (was there an isotype control?); no further mechanistic experiments are performed using human cells. This may be an issue because there are some notable differences in immunometabolic pathways seen in human versus murine material. Specifically, after stimulation, murine cells undergo Warburg metabolism, whereas human myeloid cells are more likely to increase both glycolysis and oxidative phosphorylation after a while.

Response: As suggested by the Reviewer, we have significantly strengthened the manuscript by performing several additional experiments, including the isotype controls for LDHA immunostaining in human tissue (**Figure S2**) and performing new experiments investigating immunometabolism in primary human monocyte-derived macrophages (hMDMs; **Figure S6E-G**). As noted in the revised manuscript (lines 376 – 383; lines 402 – 406), much of our work initially done with murine cells was reproduced in hMDMs. We note, however, that hMDMs exhibit distinct biology from murine macrophages. We found that in hMDMs the NAD(H) abundance and glycolytic capacity were negatively associated with the multiplicity of infection (MOI). Interestingly, the ability of NAM to restore NAD(H) abundance, preserve glycolytic capacity, and reduce the bacterial burden were also dependent on MOI. This more-complex relationship appropriately reflects known differences in the biology of humans and mice, as well as outbred and inbred populations. When considered in light of the more controlled studies in murine cells, we feel these results significantly improve the validity of our manuscript and thank the Reviewer for highlighting these issues.

As the authors know, pathways of immunity to TB are different in human and mouse cells also. Mouse cells efficiently kill Mycobacterium tuberculosis after stimulation with interferon gamma and the production of nitric oxide. Such pathways are not evident in human cells. Palmiere et al (PMID 32019928) suggested that NO-driven reprogramming of metabolism should be considered a result rather than a mediator of inflammatory polarization – in murine cells. So, have they got any macrophage data using human cells?

Response: We fully agree with the Reviewer that pathways of immunity to TB are different in human and mouse cells. To determine the optimal conditions for the Reviewer-requested experiments in **Figure S6E-G** of the revised manuscript, we performed additional experiments to first address this issue. We found that differentiation with GM-CSF but not M-CSF induced macrophages with robust glycolytic capacity (**Figure R1A-C**). Notably, this was distinct from BMDMs, which required stimulation with IFN γ to achieve a robust glycolytic capacity (**Figure 4A, B**). IFN γ stimulation of hMDMs differentiated with GM-CSF resulted in variable responses between biological replicates—ranging from no change in glycolytic capacity (**Figure R1B**) to an increase of ~30% (**Figure R1C**). We therefore omitted IFN γ from further experiments using hMDMs.

Thus, our central hypothesis that NADH homeostasis supports glycolysis and bacterial control is supported by results in human macrophages, even in the absence of IFN γ and the related pathways in murine macrophages. However, given the complexity of the figures in the manuscript, we have chosen to include these data within this response rather than in the manuscript. Instead, in the revised manuscript, we have incorporated a discussion of our findings in the context of the similarities and differences between human and murine immune cells with specific reference to Palmiere et al (lines 469 – 474). Regardless, if the Reviewer feels including the data shown in Figure R1 would substantially improve the manuscript, we can incorporate them in a revised manuscript.

Figure R1. Bar graphs representing the glycolytic capacity of hMDMs differentiated with (A) 40 ng/mL M-CSF or (B, C) 20 ng/mL GM-CSF, as determined by extracellular flux analysis. Panels (B) and (C) represent data obtained from hMDMs collected from different donors. IFN γ stimulation and extracellular flux analysis were performed as described in Materials and Methods section of the revised manuscript. Data are presented as individual values for technical replicates with the group mean \pm SEM ($n = 5$ / group) and each panel represents an independent experiment. Statistical significance was determined by two-sample t-test without assuming equal variance (B, C). *** $p < 0.001$.

The figures are complex but clear. Figures three in particular emphasizes the importance of early events in the mouse infection which dictate the final trajectory of the animal's health. In Fig 4, they use a LDH inhibitor which should be defined. In figure 5 they use a NAMP inhibitor which should be defined. In figure seven they demonstrate how NAM protects the host.

Response: We appreciate the Reviewer's close reading of our manuscript. We have incorporated the suggested changes to **Figure 4** (Lines 1096-1097) and **Figure 5** (Line 1135) in the revised manuscript.

Maybe .. they can advise us on mechanism? in other words, how is NAD+ homeostasis linked to the ability of the macrophage to kill Mtb? Is it related to NO in mice? Autophagy or phagolysosome maturation events?

Response: As suggested by the Reviewer, we have incorporated a discussion to advise the readers of the likely mechanisms downstream of the various changes to host metabolism presented in our manuscript (lines 477 – 489). This is based on data included in our initial submission and data added based on Reviewer feedback.

Reviewer #2 (Remarks to the Author):

The manuscript " NAD(H) homeostasis is essential for host protection mediated by glycolytic myeloid cells in tuberculosis"

This manuscript provides important insight into immunometabolic response of the host to Mycobacterium tuberculosis and specifically the role of glycolysis and the Warburg effect. Whether Mtb infection induces glycolysis has been confounded in the literature with studies that have found that it both enhances or represses glycolysis due to the factors discussed in the manuscript and differences in the model systems used. Here the authors data indicates that whilst Mtb does induce glycolysis the additional IFN γ stimulation of glycolysis is dampened by M. tuberculosis infection. This is shown with extensive experimental data from lungs resected from human TB patients, murine in vivo and ex vivo models of TB. The authors then explore the mechanisms behind this demonstrating that this is associated with NAD(H) homeostasis and go on to show that nicotinamide can be used as a host directed therapy. Whilst I have some minor issues this is an excellent study which is an important contribution to the field.

Response: We thank the Reviewer for their positive comments regarding our manuscript, especially highlighting the integration of studies on tissue from human TB lungs, the murine model of TB, and primary cells from mice and humans.

Specific comments

¹³C isotopologue profiling is performed within this study but the isotopologue profiles appear to be almost completely ignored, not described or discussed. This is important data and would be interesting to see also what happens to the rest of metabolism as these pathways don't act in silo.

The authors discuss differences in pool sizes to infer metabolic flux changes. Pool sizes cannot be used to do this, and this must be mentioned. The isotopologue data shows something slightly more complicated and interesting is going on with metabolism. In some of the labelling patterns whilst there is changes in pool size the isotopologue profile is unchanged indicating there is no change in the flux through the pathway. Indeed for some metabolites there is an increase in labelling suggesting increased flux at the top end of glycolysis with the lower end of glycolysis remaining unchanged presumably to conserve NAD. This looks like it is then driving flux through the PPP which would generate NADPH. This doesn't detract from the central message but if this data is presented it needs to be discussed.

Response: We agree with the Reviewer's interpretation of the isotopologue profiles, and we have included a discussion of these data in the Results section of the revised manuscript (lines 310 – 316). Specifically, we have adjusted the wording regarding pool sizes and metabolic flux, and we have introduced the observation that macrophages appear to be increasing flux through the PPP, potentially as an approach to circumvent the NAD⁺-dependent reaction catalyzed by GAPDH.

The authors postulate that NADH availability is a potential mechanism of the observed phenotypes. As a minor point the addition of pyruvate did not change the NAD/NADH balance in the LDH deficient cells which seems to be a slight anomaly that needs explaining. The authors say its expected but that is slightly puzzling to me. Figure 5C authors also measure pool sizes to show that there is a build up of metabolites which could allosterically inhibit enzymes which is fine as long as there is no inference about metabolic flux here. There is lots of metabolites presented but only two discussed. Is there isotopologue labelling for this experiment to show how the flux changes in these conditions?

Response: Based on the Reviewer's comments, we have made several improvements to the revised manuscript.

1. First, we have adjusted the text to better explain our interpretation of the NAD/NADH ratio (lines 298 – 302). In summary, the ratio reflects the pool size of NAD⁺ and NADH. Thus, while we might have expected an improvement in the relative abundance of NAD⁺ compared to NADH, these results do not preclude increased cycling (*i.e.*, flux) through these reactions.
2. Second, we have explicitly stated that no inference regarding flux should be made based on the metabolomics data (line 310) and instead emphasize the related extracellular flux analysis performed under identical conditions.
3. Third, we have included the isotopologue data associated with **Figure 5C** as **Figure S6C-D** and expanded our discussion of the metabolomics data to highlight relevant differences in isotopologue labelling for named intermediates throughout glycolysis, the PPP, and the TCA (lines 346 – 351).

Minor points

L93-94 “suggest the consequences of impaired glycolytic capacity in these cells” doesn't quite make sense. What are these consequences?

We agree with the Reviewer that this clause is confusing, and we removed it entirely. “Consequences” refers to the consequences of losing bacillary control, which is a consequence in and of itself and does not require the emphasis intended by the original wording.

L422-425 “However, these results might be explained by a short-term reduction in inflammation, since the authors state that the effect of FX11 in vivo could not be confirmed as the on-target inhibition of LDH, leaving open the potential for other mechanisms for the protective effect.” Perhaps reword as I don’t quite understand this.

Response: We agree with the Reviewer, and we have clarified the wording in the revised manuscript (lines 460 – 463).

Reviewer #3 (Remarks to the Author):

This manuscript describes a substantial number of experiments that investigated the importance of NAD(H) homeostasis, glycolytic myeloid cells and, specifically, lactate dehydrogenase (LDH) for control of infections with Mycobacterium tuberculosis. The manuscript is well organized and well written.

Response: We thank the Reviewer for their positive feedback, especially regarding the breadth of experimentation performed and how these experiments are organized and presented.

The focus on LDH was nicely motivated by expression analyses of LDHA (representing one of two LDH subunits) in lung tissues resected from TB patients. Analyses of mice lacking LDHA confirmed that inactivation of LDHA indeed causes a (somewhat modest) increase in susceptibility to M. tuberculosis. These findings are supported by experiments with bone marrow derived macrophages (BMDMs), which demonstrate that LDHA-deficient macrophages are unable to respond metabolically to stimulation with IFN-gamma. Unfortunately, it remains to be demonstrate how LDHA-deficiency impacts replication of Mtb in BMDMs.

Response: We appreciate this Reviewer’s concern regarding *Mtb* replication in BMDMs. Hence, we have performed and included the experiment suggested by the reviewer in the revised manuscript (**Figure 4G; lines 288 – 295**). In brief, our findings are consistent with our bioenergetic experiments. We found that LDHA deficiency had no impact on bacterial burden in unstimulated, untreated macrophages; LDHA^{-/-} BMDMs exhibited modestly increased bacterial burden in the setting of IFN γ stimulation. Treatment with pyruvate equalized the bacterial burden between the two groups, while the further addition of an LDH inhibitor (GSK 2837808A) abrogated the effect of pyruvate.

It is worth noting that pyruvate treatment led to a relative increase in bacterial burden relative to IFN γ stimulation alone; however, this is consistent with the previously described effect of pyruvate enhancing bacterial replication (PMID: 31201418). We also note that the relative reduction in bacterial burden following treatment with the LDH inhibitor (GSK2837808A) suggests the inhibitor may exhibit activity in *Mtb*. However, this does not influence the interpretation of any other experiments in the manuscript, as that inhibitor was not otherwise used in the presence of *Mtb*.

Several experiments were performed to demonstrate a link between NAD(H) homeostasis and glycolysis. That such a link exists is not surprising given the broad spectrum of enzymes that required NAD(H). What is surprising is the claim that NAD(H) homeostasis provides the means for a “highly selective manipulation of host-cell glycolytic flux”. It’s not clear to me what data substantiate this claim.

Response: We agree with the Reviewer that this argument is not well supported as previously written and thank the Reviewer for pointing this out. We have attenuated this claim and added additional context to support the new claim (lines 117 – 121) in the revised manuscript. To further clarify, we intended this statement as a logical argument. The comparison we are referring to when saying “selective” is primarily the administration of 2-DG and the use of knockout cells of HIF-1 α . These interventions have a profound impact on glycolysis, which is required not only for immunologic functions, but also for basic cellular processes. 2-DG halts glucose uptake and glucose metabolism entirely and significantly impairs the PPP, while HIF-1 α regulates the expression of enzymes that are related to glycolysis both indirectly, such as LDHA and NAMPT (manipulated in independently in this manuscript), and directly, such as GAPDH, PGK1, PGAM1, ALDOC (PMID: 19491311). Thus, the restriction of glycolytic capacity by disrupting NAD(H) homeostasis is a more limited and “focused” manipulation of glycolysis than the previously used approaches.

Finally, the authors demonstrate that nicotinamide (NAM) more potently inhibits growth of Mtb in BMDMs compared to standard liquid media and that oral NAM supplementation reduces the burden of Mtb in mice. These data confirm previous experiments and are not particularly surprising given the clinical data that were generated decades ago (see PMID 12567303 for a summary). However, I agree with the authors that NAM supplementation might be of clinical value in specific circumstances.

Response: We agree with the Reviewer; some of the experiments presented confirm previous historical work. By replicating the original findings with modern techniques, we add value to the literature by (i) reintroducing nicotinamide as a TB therapy to a modern audience and (ii) provide data on the efficacy of NAM with modern techniques so that it may be compared against other proposed host-directed therapies. We further identify pathways in host cells on which the activity of NAM depends.

A central claim of the manuscript is that the impact of NAM supplementation is mediated by a “host-directed antimicrobial effect”. Unfortunately, this claim is only weakly supported by experimental data. This claim is also at odds with a previously report that failed to detect an impact of NAM on macrophages infected with Mtb containing a deletion in pncA, which renders Mtb resistant to the direct antibacterial effects of NAM. This report (PMID 31665359) is cited but the discrepancy in conclusions is not discussed and it’s surprising that the authors did not utilize NAM resistant MTB to support one of their main conclusions.

Response: We regret the lack of clarity on these issues. The Reviewer highlights two distinct concerns in our manuscript: (i) a purported discrepancy with previous work (PMID 31665359) and (ii) weak experimental support for the claim that NAM exerts a host-directed antimicrobial effect that would be enhanced with the *Mtb*-pncAdel mutant utilized in PMID 31665359.

1. With regard to the first point, we agree with the reviewer that we have not adequately discussed our findings in the context of the article in question (PMID 31665359). We have now included a more thorough discussion relating our study with theirs (lines 392 – 394; lines 501 – 506) in the revised manuscript. We do, however, respectfully disagree with the Reviewer’s interpretation of the article in question, because the authors of this article conclude that their work “*suggests host-specific NAM targets rather than PncA-dependent direct antimicrobial properties,*” arguing that “*host pathways may be targeted by NAM to restrict BCG*

growth...”—which lacks a functional PncA—“...and contribute to macrophage control of M. tuberculosis replication.”

Regarding the specific point of the inefficacy of NAM in their *in vitro* infection model utilizing virulent *Mtb-pncA^{del}*, the authors suggest that the limited viability of macrophages within the constraints of their system may preclude observation of the host-directed effect seen in BCG at later time points. Thus, our revised manuscript reflects this interpretation of their findings.

2. We agree with the Reviewer that the claim NAM exerts a “host-directed antimicrobial effect” is not fully substantiated in our manuscript. Hence, we clarify in the revised manuscript that NAM exerts a “host-dependent” effect. It is our view that this claim is strongly supported with the following lines of evidence: (i) The efficacy of NAM is dependent on the presence of macrophages (**Fig 6A**). (ii) Two inhibitors that enhance the growth of *Mtb* in macrophages disrupt the activity of NAM but not the activity of the direct-acting antimycobacterial control drug, isoniazid (**Fig 6B, C**). (iii) Included in the revised manuscript, the effect of NAM in human macrophages is negatively associated with the ratio of bacteria to macrophages (**Figure S6**). Thus, the efficacy of NAM depends on the very pathways it supports (NAD⁺ salvage and glycolysis), regardless of downstream PncA-dependent or PncA-independent mechanisms. To address this concern, we have replaced “host-directed” with “host-dependent” throughout the manuscript where appropriate (*i.e.*, with the exception of our initial hypothesis on lines 396-397), and we have included a thorough discussion addressing that the inability to distinguish host-directed versus host-dependent is a limitation of our work (lines 501 – 512).

REVIEWERS' COMMENTS

Reviewer #1 (Remarks to the Author):

The negative control data and human macrophage experiments are noted, as requested.

Reviewer #2 (Remarks to the Author):

I am happy that all my concerns and those of the other reviewers have been sufficiently addressed. This work presents a very interesting addition to the literature.